# 3DMolFormer: A Dual-channel Framework for Structure-based Drug Discovery

**Xiuyuan Hu**[1], **Guoqing Liu**[2*], **Can (Sam) Chen**[3,4], **Yang Zhao**[1], **Hao Zhang**[1*], **Xue Liu**[3,4]

[1]Department of Electronic Engineering, Tsinghua University, [2]Microsoft Research AI for Science
[3]McGill University, [4]Mila - Quebec AI Institute

`huxy22@mails.tsinghua.edu.cn`, `guoqingliu@microsoft.com`,
`can.chen@mila.quebec`, `zhao-yang@tsinghua.edu.cn`,
`haozhang@tsinghua.edu.cn`, `xueliu@cs.mcgill.ca`

## Abstract

Structure-based drug discovery, encompassing the tasks of protein-ligand docking and pocket-aware 3D drug design, represents a core challenge in drug discovery. However, no existing work can deal with both tasks to effectively leverage the duality between them, and current methods for each task are hindered by challenges in modeling 3D information and the limitations of available data. To address these issues, we propose 3DMolFormer, a unified dual-channel transformer-based framework applicable to both docking and 3D drug design tasks, which exploits their duality by utilizing docking functionalities within the drug design process. Specifically, we represent 3D pocket-ligand complexes using parallel sequences of discrete tokens and continuous numbers, and we design a corresponding dual-channel transformer model to handle this format, thereby overcoming the challenges of 3D information modeling. Additionally, we alleviate data limitations through large-scale pre-training on a mixed dataset, followed by supervised and reinforcement learning fine-tuning techniques respectively tailored for the two tasks. Experimental results demonstrate that 3DMolFormer outperforms previous approaches in both protein-ligand docking and pocket-aware 3D drug design, highlighting its promising application in structure-based drug discovery. The code is available at: `https://github.com/HXYfighter/3DMolFormer`.

## 1 Introduction

In recent years, the application of machine learning in drug discovery has gained significant traction (Mak et al., 2023), achieving substantial advancements in tasks such as molecular property prediction (Zhang et al., 2021; Wang et al., 2021; Wieder et al., 2020), protein structure prediction (Jumper et al., 2021; Baek et al., 2021; Lin et al., 2023), and drug molecular design (Olivecrona et al., 2017; Luo et al., 2021; Fu et al., 2022; Du et al., 2022; 2024). These developments hold the promise of dramatically enhancing the efficiency of drug development processes (Blanco-Gonzalez et al., 2023). Notably, the transformer architecture, which has seen breakthroughs in natural language processing (NLP) (Devlin et al., 2019; Brown et al., 2020), has been successfully adapted for molecular representation learning (Zhou et al., 2023a; Gao et al., 2024a), protein-ligand interaction prediction (Zhao et al., 2022; Abramson et al., 2024), and molecular generation tasks (Bagal et al., 2021; Hu et al., 2023).

Structure-based drug discovery (SBDD) is one of the most critical strategies in drug discovery practices, relying on theories of drug-receptor interactions to study the complexes formed between protein pockets and small molecule ligands (Van Montfort & Workman, 2017). SBDD encompasses two core tasks: (1) protein-ligand binding pose prediction (docking), which involves predicting the 3D binding conformation of a ligand given the 3D structure of a protein and the 2D representation of the ligand (Yang et al., 2022), and (2) pocket-aware 3D drug design, which entails designing 3D drug molecules that bind well (with low binding energy) to a given pocket target on a protein

---

*Corresponding authors: Hao Zhang (1st), Guoqing Liu.

structure (Zhang et al., 2023b; Isert et al., 2023b). These two tasks are inherently dual, and one is predictive, while the other is generative.

However, as of now, the application of machine learning in these two SBDD tasks remains widely recognized as a challenge (Pala & Clark, 2024). The accuracy and generalization of protein-ligand docking methods are still unsatisfactory (Morehead et al., 2024), and pocket-aware 3D drug design approaches have not achieved obvious improvements by explicitly utilizing 3D structural information compared to 2D methods (Zheng et al., 2024). This predicament can be attributed to three primary reasons:

- **Underutilized duality**: Protein-ligand docking and pocket-aware 3D drug design are naturally dual tasks, and improvements in docking performance could directly benefit drug design. However, since these two tasks are different in type (predictive vs. generative), this duality has unfortunately not been leveraged by previous machine learning approaches.

- **Challenges in modeling 3D information**: Modeling 3D information is a key difficulty in SBDD, as protein sequences and small molecule graphs contain only discrete information, whereas 3D coordinates are continuous values. Merging these two modalities of information has proven challenging (Zhu et al., 2022).

- **Limited data**: Ground-truth data on protein-ligand complexes are scarce. Currently, the largest dataset, PDBbind (Liu et al., 2017), contains fewer than 20,000 complexes, which is insufficient for training a robust machine learning model.

To address these challenges, we propose 3DMolFormer, a unified transformer-based framework for both of the two SBDD tasks. First, to fulfill the input-output causal relationships essential for both docking and 3D drug design, we introduce a parallel sequence format to represent a 3D complex of a protein pocket and a small molecule ligand, as shown in Figure 1 and 2, which comprises a token sequence for discrete protein atoms and small molecule SMILES, alongside a numerical sequence for 3D coordinates. Subsequently, we construct the 3DMolFormer model based on this parallel sequence, as illustrated in Figure 3, augmenting the GPT architecture (Radford et al., 2019) with a numerical head corresponding to the token head, enabling the model to be directly applied for autoregressive generation of the parallel sequences.

Due to data limitations, we utilize a "pre-training + fine-tuning" approach (Quinn et al., 2019) in NLP for 3DMolFormer, as large-scale pre-training helps mitigate these data challenges. During the pre-training phase, the model undergoes large-batch training (Keskar et al., 2017) on a large-scale mixed dataset, which includes data on protein pockets, ligands, and pocket-ligand complexes. A composite loss function is employed for autoregressive training of the parallel sequence, where cross-entropy loss applies to the token sequence and mean squared error loss applies to the numerical sequence. For the protein-ligand docking task, we perform supervised fine-tuning on the ground-truth pocket-ligand complexes, using the mean squared error of the numerical sequences corresponding to the ligand's 3D coordinates as the loss function. Moreover, to utilize the duality between the two SBDD tasks, for the pocket-aware drug design task, we apply a regularized maximum likelihood estimation loss for reinforcement learning fine-tuning, and leverage the weights fine-tuned for docking to generate the 3D coordinates of the small molecules.

Experimental results for protein-ligand docking demonstrate that 3DMolFormer outperforms all search-based and deep-learning docking baselines in binding pose prediction accuracy, particularly showing a reduction in samples with large prediction errors. Results for pocket-aware 3D drug design indicate that through a carefully designed composite reward function, 3DMolFormer can generate drug candidates that meet satisfactory levels of binding affinity (docking score), drug-likeness, and synthesizability during the reinforcement learning process, in particular significantly surpassing existing state-of-the-art baselines in terms of binding affinity and success rates in meeting multi-objective criteria. These results reflect the outstanding performance of the 3DMolFormer framework in structure-based drug discovery.

In summary, our main contributions include:

- We propose 3DMolFormer, the first unified framework applicable to both protein-ligand docking and pocket-aware 3D drug design.

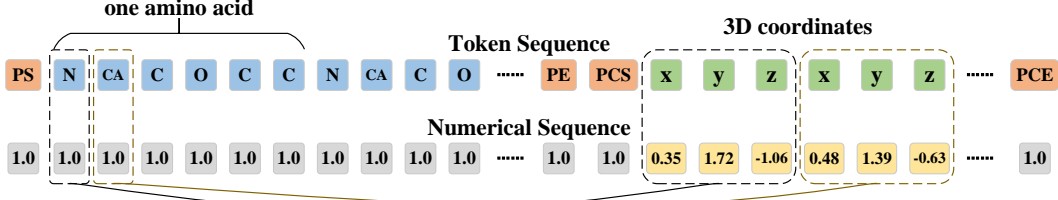

Figure 1: The parallel sequence of a protein pocket with 3D coordinates.

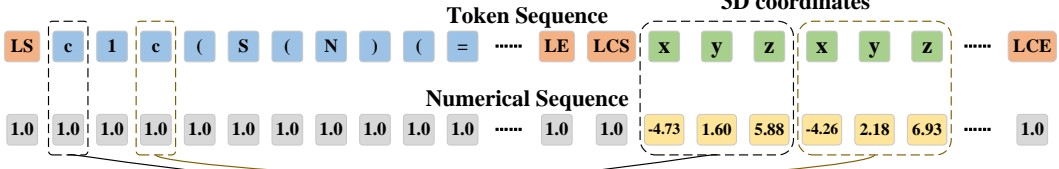

Figure 2: The parallel sequence of a small molecule ligand with 3D coordinates.

- We design a parallel sequence format for pocket-ligand complexes and establish a dual-channel transformer architecture to autoregressively generate this format, effectively addressing the challenges of modeling 3D information in SBDD.

- Through large-scale pre-training and respective fine-tuning, 3DMolFormer outperforms various previous baselines in both SBDD tasks.

## 2 RELATED WORKS

**Molecular Pre-training** The success of large-scale pre-training has extended from NLP to the field of drug discovery (Xia et al., 2023; Chen et al., 2023). Many studies focus on molecular representation learning, which maps molecular structures to informative embeddings for downstream predictive tasks Yang et al. (2021a); Fang et al. (2022); Gao et al. (2024a). Several representation learning methods for protein-ligand binding have been proposed, including InteractionGraph-Net (Jiang et al., 2021) and BindNet (Feng et al., 2024a), with Uni-Mol (Zhou et al., 2023a) collecting and pre-training on extensive 3D datasets of proteins and small molecules, achieving high accuracy in protein-ligand docking. Furthermore, models such as MolGPT (Bagal et al., 2021), Chemformer (Irwin et al., 2022), and BindGPT (Zholus et al., 2024) utilize pre-training to enhance molecular distribution learning, enabling applications in generative tasks.

**Protein-ligand Docking** Protein-ligand docking encompasses three sequential tasks: binding site prediction, binding pose prediction, and binding affinity prediction, with binding pose prediction being the most critical in structure-based drug discovery (Zhang et al., 2023b). Traditional search-based methods typically employ combinatorial optimization techniques to identify the best binding poses (known as targeted docking) within a given protein pocket, using tools such as AutoDock4 (Morris et al., 2009), AutoDock Vina (Trott & Olson, 2010; Eberhardt et al., 2021), and Smina (Koes et al., 2013), which are widely used in practical virtual screening. Recently, deep learning approaches have been introduced for this task, exemplified by DeepDock (Méndez-Lucio et al., 2021) and Uni-Mol (Zhou et al., 2023a). Additionally, various deep learning techniques for blind docking have emerged, which simultaneously predict binding sites and poses (Stärk et al., 2022; Lu et al., 2022; Zhang et al., 2023a; Pei et al., 2024; Corso et al., 2023; 2024). However, blind docking methods are primarily hindered by inaccuracies in binding site prediction, making direct comparisons with targeted docking methods less meaningful. Moreover, some end-to-end approaches that predict binding affinity without 3D poses fail to provide the crucial structural information required in SBDD (Wang et al., 2024a).

**Pocket-aware 3D Drug Design** Drug design is the ultimate goal of molecular design. Currently, most machine learning methods focus on generating 1D SMILES strings or 2D molecular graphs (Segler et al., 2018; Eckmann et al., 2022; Lee et al., 2024), with reinforcement learning

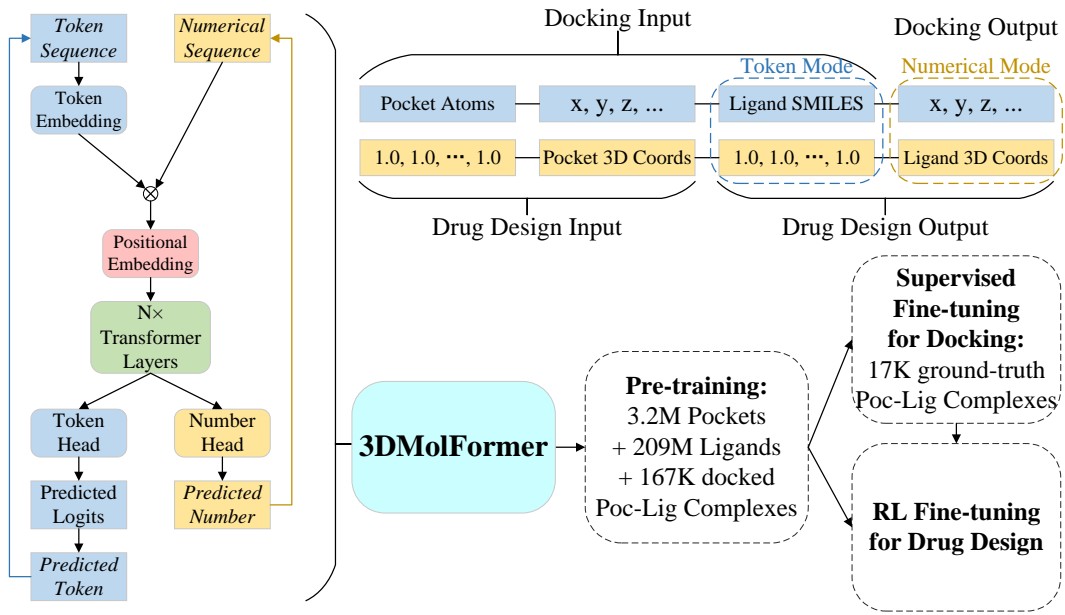

Figure 3: Overview of 3DMolFormer. The left shows the dual-channel model architecture, the top right illustrates the input and output of the two SBDD tasks in a parallel sequence, and the bottom right outlines the pre-training and fine-tuning process.

being a popular paradigm (Olivecrona et al., 2017; You et al., 2018; Ahn et al., 2020; Jin et al., 2020; Simm et al., 2020; Yang et al., 2021b). However, these approaches can only output discrete information about atoms and chemical bonds, lacking the capability to generate 3D coordinate values, thus limiting their application in SBDD. In contrast, pocket-aware 3D drug design explicitly utilizes the 3D structures of protein targets to generate *de novo* small molecules with high binding affinity. Various machine learning techniques have been applied to pocket-aware 3D drug design, including genetic algorithms (e.g., AutoGrow (Spiegel & Durrant, 2020)), variational autoencoders (e.g., liGAN (Ragoza et al., 2022)), autoregressive models (e.g., AR (Luo et al., 2021), Pocket2Mol (Peng et al., 2022), Lingo3DMol (Feng et al., 2024b)), and flow models (GraphBP (Liu et al., 2022)). Recently, diffusion models have achieved state-of-the-art performance in this task, including DiffSBDD (Schneuing et al., 2022), TargetDiff (Guan et al., 2023a), and DecompDiff (Guan et al., 2023b). Notably, some studies have developed transformer-based 3D drug design models. The XYZ-transformer (Flam-Shepherd & Aspuru-Guzik, 2023) directly uses 3D coordinate values (retaining three decimal places) as tokens, while BindGPT (Zholus et al., 2024) decomposes the integer and decimal parts of coordinates into two tokens to reduce vocabulary size. Token-Mol (Wang et al., 2024b), on the other hand, employs torsion angles of small molecules instead of coordinate values to shorten sequence lengths. However, these methods represent values using discrete tokens, which disrupts the continuity of coordinates.

## 3   3DMOLFORMER

### 3.1   FORMAT OF POCKET AND LIGAND SEQUENCES WITH 3D COORDINATES

To leverage a causal language model for handling 3D protein pockets and small molecules while explicitly separating discrete structural information from continuous spatial coordinates, we design a parallel sequence format. This format consists of a discrete token sequence $s_{\text{tok}}$ and a continuous numerical sequence $s_{\text{num}}$, both of which share the same length and align element-wise. The token sequence consists of tokens in a predefined vocabulary, while the numerical sequence contains floating-point values.

As shown in Figure 1, the sequence for a protein pocket $s^{\text{poc}}$ consists of two parts: the first $s^{\text{poc\_atoms}}$ represents an atomic list, and the second $s^{\text{poc\_coord}}$ contains 3D coordinate information. The atomic list is encoded in the token sequence, which includes all atoms in the protein pocket except for

hydrogen atoms. Aside from alpha carbon atoms, denoted as 'CA', other atoms are represented by their element type, such as 'C', 'O', 'N', and 'S'. The sequence of atoms follows the order of the pdb file, where each amino acid begins with ['N', 'CA', 'C', 'O'] followed by the side-chain atoms. The normalized 3D coordinates for each atom in the atomic list are included in the numerical sequence in the same order, with each dimension ('x', 'y', 'z') occupying a separate position. The length of the 3D coordinate sequence is always three times the length of the atomic list. Moreover, in the token sequence, the start and end of the atomic list are marked by 'PS' and 'PE', while the 3D coordinates are delineated by 'PCS' and 'PCE' at the start and end, respectively. In the numerical sequence, numbers that do not correspond to 3D coordinates are padded with 1.0.

As illustrated in Figure 2, the sequence for a small molecule $s^{\text{lig}}$ is similar to that of the protein pocket, comprising both a SMILES string section $s^{\text{lig-smiles}}$ and a 3D coordinate section $s^{\text{lig-coord}}$. After atom-level tokenization (Schwaller et al., 2019), the SMILES string of the small molecule is encoded in the token sequence, excluding hydrogen atoms. It is important to note that some tokens may not correspond to atoms, and thus, no 3D coordinates will be associated with them. The normalized 3D coordinates for each atom in the tokenized SMILES string are included in the numerical sequence, with each coordinate dimension ('x', 'y', 'z') occupying a separate position. The length of the 3D coordinate sequence is always three times the number of atoms in the small molecule. In the token sequence, the start and end of the SMILES tokens are marked by 'LS' and 'LE', while the 3D coordinates of the corresponding atoms are marked by 'LCS' and 'LCE' at the start and end, respectively. In the numerical sequence, numbers not corresponding to 3D coordinates are similarly padded with 1.0.

When the sequence of a protein pocket is concatenated with that of a small molecule ligand, it forms a pocket-ligand complex sequence along with their 3D coordinates $s^{\text{poc}-\text{lig}}$. This sequence format offers three advantages:

- It fully encapsulates the structural and 3D coordinate information of both the protein pocket and the small molecule ligand.

- Discrete structural information and continuous numerical data are separated into two parallel sequences, enabling independent processing of each data type.

- The sequence of the pocket-ligand complex maintains causal logic. As depicted in the upper right of Figure 3, this sequence structure allows autoregressive prediction, which can effectively represent both pocket-ligand docking and pocket-aware drug design tasks.

Specifically, we normalize the coordinates of all pocket-ligand complexes by translating their center of mass to the origin $(0, 0, 0)$. Additionally, to ensure numerical stability during training (Quinn et al., 2019), we scale the coordinate values by a factor $q > 1$ to reduce the range of their distribution:

$$(x'_i, y'_i, z'_i) = \left( \frac{x_i - x_c}{q}, \frac{y_i - y_c}{q}, \frac{z_i - z_c}{q} \right), \tag{1}$$

where $(x_i, y_i, z_i)$ is the original coordinate of the $i$-th atom, $(x_c, y_c, z_c)$ is the coordinate of the center of mass, and $(x'_i, y'_i, z'_i)$ refers to the normalized values used in the numerical sequence.

## 3.2 MODEL ARCHITECTURE

To process the aforementioned parallel sequences, we require an autoregressive language model that can simultaneously take a discrete token sequence and a continuous floating-point sequence as input, while predicting both the next token and the next numerical value. Inspired by xVal (Golkar et al., 2023), we propose a dual-channel transformer architecture for 3DMolFormer, as illustrated in the left part of Figure 3. The module handling the token sequence is based on the GPT-2 model (Radford et al., 2019), featuring identical token embeddings, positional embeddings, multiple transformer layers, and a prediction head for logits. On top of this, we introduce a parallel numerical channel at both the input and output stages.

At the input stage, we multiply the embedding of each token in the token sequence with the corresponding value in the numerical sequence, using this product as the input to the positional embedding. This is why numerical values that lack meaningful information are padded with 1.0. At the output stage, in parallel with the token prediction head, we add a number head to predict the next floating-point value.

During inference with 3DMolFormer, the outputs are handled in two modes:

- Token Mode: In the drug design task, when predicting ligand SMILES tokens, the corresponding numerical output holds no meaningful value and is therefore padded with 1.0.

- Numerical Mode: In docking and drug design tasks, once the ligand SMILES is determined, the length of the 3D coordinate sequence and its tokens are also fixed. Therefore, the token output no longer holds meaningful information and is filled with the expected tokens (from ['x', 'y', 'z', 'LCS', 'LCE']). When the position corresponds to ['x', 'y', 'z'], the predicted floating-point values are appended to the input numerical sequence. For tokens corresponding to ['LCS', 'LCE'], the numerical values are also set to 1.0.

## 3.3 SELF-SUPERVISED PRE-TRAINING

To enable the 3DMolFormer model to learn the general patterns of pocket-ligand complex sequences, we conduct large-scale pre-training on 3D data, which includes three datasets: approximately 3.2M protein pockets, about 209M small molecule conformations, and around 167K pocket-ligand complexes. The first two datasets were collected by Uni-Mol (Zhou et al., 2023a) for large-scale pre-training on 3D protein pockets and small molecules, while the last dataset was generated by CrossDocked2020 (Francoeur et al., 2020).

In order for the dual-channel autoregressive model to capture both the token sequence format and the 3D coordinate patterns of pocket-ligand complexes, we adopt a composite loss function for the prediction of the next token and the corresponding numerical value. This loss function incorporates the cross-entropy (CE) loss for the whole token sequence and the mean squared error (MSE) loss for the numerical sequence corresponding to the 3D coordinates:

$$L(\hat{s}, s) = \text{CE}(\hat{s}_{\text{tok}}, s_{\text{tok}}) + \alpha \cdot \text{MSE}(\hat{s}_{\text{num}}^{\text{coord}}, s_{\text{num}}^{\text{coord}}),  \tag{2}$$

where $\hat{s}$ represents the sequence predicted by 3DMolFormer, $s$ refers to the training data, and $\alpha$ is a coefficient that controls the balance between the CE loss and the MSE loss. This composite loss is applied to all of the three types of pre-training data.

Additionally, we employ large-batch training (Keskar et al., 2017) through gradient accumulation, which we found to be crucial for the pre-training stability of 3DMolFormer. For further details on pre-training and hyper-parameter settings, please refer to Section 4 and Appendix B.

## 3.4 FINE-TUNING

After the large-scale pre-training, we further fine-tune the 3DMolFormer model on two downstream drug discovery tasks: supervised fine-tuning for pocket-ligand docking, and reinforcement learning (RL) fine-tuning for pocket-aware drug design.

### 3.4.1 SUPERVISED FINE-TUNING FOR PROTEIN-LIGAND BINDING POSE PREDICTION

In the protein-ligand binding pose prediction (docking) task, as illustrated in Figure 3, each sample consists of a pocket-ligand complex. The input sequence contains the atoms of the protein pocket and their 3D coordinates, along with the SMILES sequence of the ligand. The output is the 3D coordinates of each atom in the ligand.

The pre-training data for 3DMolFormer already includes about 167K pocket-ligand complexes from CrossDocked2020 (Francoeur et al., 2020); however, these complexes are generated using the docking software Smina Koes et al. (2013), which means that the docking performance of models trained with this data would not exceed that of Smina. To improve the upper limit of our model's docking performance, we fine-tune it on the experimentally determined PDBBind dataset Liu et al. (2017), which contains approximately 17K ground-truth pocket-ligand complexes. Additionally, we employ a task-specific loss function that computes the mean squared error (MSE) loss only for the 3D coordinates of the ligand in the context of next numerical value prediction, since the inference process of docking operates entirely in numerical mode:

$$L_{\text{docking}}(\hat{s}^{\text{lig-coord}}, s^{\text{lig-coord}}) = \text{MSE}(\hat{s}_{\text{num}}^{\text{lig-coord}}, s_{\text{num}}^{\text{lig-coord}}).  \tag{3}$$

To mitigate overfitting during supervised fine-tuning, SMILES randomization (Arús-Pous et al., 2019) and random rotation of the 3D coordinates of complexes are used as data augmentation strategies. For further details on docking fine-tuning, please refer to Section 4.1 and Appendix C.

### 3.4.2 RL Fine-tuning for Pocket-aware 3D Drug Design

In the pocket-aware drug design task, as illustrated in Figure 3, each sample is also a pocket-ligand pair. The input sequence includes the atoms of the protein pocket and their 3D coordinates, while the output consists of the ligand SMILES sequence and the 3D coordinates of its atoms.

Inspired by 1D RL-based molecular generation methods (Olivecrona et al., 2017), an RL agent with the 3DMolFormer architecture is initialized with the pre-trained weights, and a molecular property scoring function for each protein pocket is designed as the RL reward. Then, the agent is iteratively optimized to maximize the expected reward of its outputs. Specifically, at each RL step, the agent samples a batch of 3D ligands, and the regularized maximum likelihood estimation (MLE) loss (Svensson et al., 2023) of each ligand is computed and used to update the agent:

$$L_{\text{design}}(\hat{s}^{\text{lig}}) = \left( \log \pi_{\text{pre-trained}}(\hat{s}^{\text{lig\_smiles}}_{\text{tok}}) + \sigma \cdot R(m) - \log \pi_{\text{agent}}(\hat{s}^{\text{lig\_smiles}}_{\text{tok}}) \right)^2, \tag{4}$$

where $\hat{s}^{\text{lig}}$ ($\hat{s}^{\text{lig\_smiles}}$ and $\hat{s}^{\text{lig\_coord}}$) is a sample generated by the RL agent, $m$ is the 3D molecule represented by $\hat{s}^{\text{lig}}$, and $R(\cdot)$ is reward function evaluating the property of the molecule. $\pi_{\text{pre-trained}}(s)$ is the likelihood of the pre-trained 3DMolFormer model for generating the sequence $s$, $\pi_{\text{agent}}(s)$ is the corresponding likelihood of the agent model, and $\sigma$ is a coefficient hyper-parameter to control the importance of the reward. This loss function encourages the agent to generate molecules with higher expected rewards while retaining a low deviation from the pre-trained weights.

It is important to note that to leverage the duality of the two SBDD tasks, the sampling of ligand SMILES utilizes the weights of the RL agent's model, which are continuously updated during fine-tuning. In contrast, the generation of atomic 3D coordinates uses the weights from the model fine-tuned for docking, which remains unchanged during this process. For additional details on RL fine-tuning and hyper-parameter settings, please refer to Section 4.2 and Appendix D.

## 4 Experiments

In this section, we present the results of two parts of experiments: pocket-ligand docking and pocket-aware 3D drug design. Through pre-training, the 3DMolFormer model is theoretically capable of being applied to the conformation generation of small molecules. However, as Zhou et al. (2023b) pointed out, the existing benchmarks for conformation generation are wrong; therefore, this experiment is not conducted.

Following the configuration of the GPT-2 small model (Radford et al., 2019), the 3DMolFormer model with a total of 92M parameters has 12 transformer layers, each containing 12 self-attention heads, and the embedding dimension is 768. The maximum length for the parallel sequences is set to 2048, which covers over 99% of the samples in the training set as well as all samples in the test set for protein-ligand docking.

For pre-training, all samples with a coordinate range larger than 40 are screened out. Then, we replicate each protein pocket five times and each pocket-ligand complex twenty times, mixing them with small molecule conformations, resulting in a total of 228M training data samples. 3DMolFormer is pre-trained on this dataset for only one epoch, using a batch size of 10K implemented by gradient accumulation. The maximal learning rate is set to $5 \times 10^{-4}$ with a warmup period of 1% steps followed by cosine decay. An AdamW optimizer Loshchilov & Hutter (2019) with a weight decay factor of 0.1 is employed, and the coefficient $\alpha$ in the loss function of Eq. (2) is set to 1.0. The pre-training process takes less than 48 hours with 4 A100 80G GPUs. For further details on the selection of hyper-parameters for pre-training, please refer to Appendix B.

### 4.1 Protein-ligand Binding Pose Prediction

Experiments of protein-ligand binding pose prediction are conducted in the targeted and semi-flexible docking scenario, where the protein pocket for binding is specified and fixed, while the ligand conformation is entirely flexible.

Table 1: Experimental results of 3DMolFormer, its variants, and other baselines on protein-ligand binding pose prediction, following the results reported in Uni-Mol (Zhou et al., 2023a). (↑) / (↓) denotes that a higher / lower value is better. The best result in each column is **bolded**.

| Methods | %<1.0Å (↑) | %<2.0Å (↑) | %<3.0Å (↑) | %<5.0Å (↑) | Avg. (↓) |
|---|---|---|---|---|---|
| AutoDock4 | 21.8 | 35.4 | 47.0 | 64.6 | 3.53 |
| AutoDock Vina | 44.2 | 64.6 | 73.7 | 84.6 | 2.37 |
| Vinardo | 41.8 | 62.8 | 69.8 | 76.8 | 2.49 |
| Smina | **47.4** | 65.3 | 74.4 | 82.1 | 1.84 |
| Uni-Mol | 43.2 | 80.4 | 87.0 | 94.0 | 1.62 |
| 3DMolFormer w/o PT | 15.5 | 57.8 | 78.1 | 92.4 | 2.25 |
| 3DMolFormer w/o DA | 10.3 | 51.0 | 74.9 | 91.6 | 2.45 |
| 3DMolFormer | 43.8 | **84.9** | **96.4** | **98.8** | **1.29** |

**Data**   Following Uni-Mol (Zhou et al., 2023a), we use PDBbind v2020 (Liu et al., 2017) as the training set for supervised fine-tuning on protein-ligand docking and CASF-2016 (Su et al., 2018) as the test set, which includes 285 test samples. In addition, we apply the same data filtering process as Uni-Mol to remove training samples with high similarity to the protein sequences or molecular structures of the complexes in the test set, which results in a training set comprising 18,404 ground-truth complexes.

**Baselines**   We select four search-based methods: AutoDock4 (Morris et al., 2009), AutoDock Vina (Trott & Olson, 2010; Eberhardt et al., 2021), Vinardo (Quiroga & Villarreal, 2016), and Smina (Koes et al., 2013), along with Uni-Mol (Zhou et al., 2023a), which is currently the state-of-the-art deep learning method for targeted docking, as our baselines.

**Ablation Studies**   Two variants of 3DMolFormer are established: (1) training a 3DMolFormer model from scratch on the fine-tuning set for protein-ligand docking without pre-training (w/o PT), and (2) fine-tuning the pre-trained 3DMolFormer model without data augmentation (w/o DA).

**Evaluation**   The root mean square deviation (RMSD) between the predicted ligand pose and the ground truth is used to assess binding pose accuracy. Specifically, two metrics are employed: (1) the percentage of RMSD results that fall below predefined thresholds, with higher percentages indicating better performance, and (2) the average RMSD, where lower values are preferred.

**Fine-tuning**   For supervised fine-tuning for pocket-ligand binding pose prediction, we train the model for 2000 epochs with a batch size of 128. The maximum learning rate is set to $1 \times 10^{-4}$, with a warmup period of 1% of the steps and cosine decay applied thereafter. The training process takes less than 24 hours with 4 A100 80G GPUs.

**Results**   As shown in Table 1, 3DMolFormer outperforms all baselines in both average RMSD and the percentage of predictions with RMSD less than 2.0, 3.0, and 5.0 Å. Notably, it significantly surpasses other methods in the percentages for RMSD below 3.0 and 5.0 Å. This indicates that 3DMolFormer is less prone to making "large errors" compared to the baselines, reflecting its robustness. However, for the percentage of predictions with RMSD below 1.0 Å, the search-based method Smina outperforms the deep learning approaches, suggesting that there is still room for improvement in the ability of deep learning methods to capture the intricate interactions between protein pockets and ligands. Moreover, the ablation studies demonstrate that the pre-training and data augmentation both play a crucial role in the training of the 3DMolFormer docking model.

It is worth noting that, unlike all baseline methods, 3DMolFormer does not require an initialized 3D conformation of the ligand as input, indicating that the model has acquired the capability to predict small molecule 3D conformations through pre-training. This feature enhances the usability of 3DMolFormer compared to previous docking approaches.

Additionally, the average time taken by 3DMolFormer to predict a binding pose is 0.8 seconds using 1 A100 80G GPU, and this can be significantly accelerated through parallel inference. This suggests that 3DMolFormer has great potential for applications in large-scale virtual screening. For

further details and results of experiments on protein-ligand binding pose prediction, please refer to Appendix C.

## 4.2 Pocket-aware 3D Drug Design

In the experiments for pocket-aware 3D drug design, small molecule ligands and their 3D conformations are designed to bind well with a specified pocket on a protein whose structure remains fixed.

**Data**  Following previous works (Peng et al., 2022; Guan et al., 2023a;b), we select 100 protein pockets from the CrossDocked2020 (Francoeur et al., 2020) dataset that exhibit low similarity ($<$ 30%) to the protein sequences of pocket-ligand complexes used in pre-training, thereby establishing our targets for 3D drug design.

**Baselines**  We compare 3DMolFormer against various baselines for pocket-aware 3D molecular generation, including AR (Luo et al., 2021), liGAN (Ragoza et al., 2022), GraphBP (Liu et al., 2022), Pocket2Mol (Peng et al., 2022), TargetDiff (Guan et al., 2023a), and DecompDiff (Guan et al., 2023b). Additionally, we report the results of the ligands corresponding to the 100 protein pockets in the CrossDocked2020 dataset for reference.

**Evaluation**  In alignment with previous works, we evaluate 100 3D molecules generated for each protein pocket. Four metrics are selected to comprehensively assess the potential of generated molecules in practical drug design: (1) **Vina Score**, which directly estimates the binding affinity based on the generated 3D molecules; (2) **Vina Dock**, representing the best possible binding affinity of the molecules estimated by re-docking; (3) **QED** (Quantitative Estimate of Drug-likeness) (Bickerton et al., 2012); and (4) **SA** (Synthetic Accessibility) (Ertl & Schuffenhauer, 2009)[1]. We employ Quick Vina 2 (Alhossary et al., 2015) to estimate the binding affinity, which is an efficient alternative to AutoDock Vina. For all metrics, we report their average values across designed drug molecules for all protein pockets. Following Long et al. (2022) and Guan et al. (2023b), we also report the percentage of designed drug molecules meeting specific criteria: Vina Dock$< -8.18$, QED $> 0.25$, and SA$> 0.59$. This percentage, referred to as the **Success Rate**, reflects the performance of different methods in multi-objective drug design, which is a common scenario in practical drug discovery.

**Reward Function**  For the aforementioned drug design objectives, we formulate a composite reward function for the RL fine-tuning process ($R(m)$ in Eq. (4)). First, a reverse sigmoid function (Hu et al., 2023) is applied to transform the Vina Dock score into a range of $[0, 1]$, where higher values are preferable:

$$R_{\text{Dock}}(m) = 1/(1 + 10^{0.625 \cdot (\text{VinaDock}(m)+10)}), \tag{5}$$

where $m$ refers to a small molecule.

Next, we utilize a step function for QED and SA, as these properties are auxiliary to the docking score; thus, they only need to exceed certain thresholds rather than aiming for higher values.

$$R_{\text{QED}}(m) = \mathbb{I}(\text{QED}(m) > 0.25), \quad R_{\text{SA}}(m) = \mathbb{I}(\text{SA}(m) > 0.59), \tag{6}$$

where $\mathbb{I}(\cdot)$ represents the indicator function.

Finally, the mean of these three scores is employed as the RL reward function:

$$R(m) = \frac{1}{3}\big[R_{\text{Dock}}(m) + R_{\text{QED}}(m) + R_{\text{SA}}(m)\big]. \tag{7}$$

This composite reward is also used as the multi-objective criteria for selecting drug candidates from all generated molecules.

**Fine-tuning**  For the reinforcement learning fine-tuning aimed at pocket-aware 3D drug design, we execute 500 RL steps for each protein pocket, with a batch size of 128 and a constant learning rate of $1 \times 10^{-4}$. The parameter $\sigma$ in Eq. (4) is set to 100. The RL process for each protein pocket takes less than 8 hours using 1 A100 80G GPU and 128 CPU cores, with the computation of the Vina Dock reward running in parallel on the CPU cores.

---

[1]Here the original SA score has been linearly transformed to $[0, 1]$, as illustrated in Appendix D.

Table 2: Experimental results of 3DMolFormer and other baselines on pocket-aware 3D drug design, following the results reported in DecompDiff (Guan et al., 2023b). (↑) / (↓) denotes that a higher / lower value is better. The best result in each column is **bolded**.

| Methods | Vina Score (↓) | Vina Dock (↓) | QED (↑) | SA (↑) | Success Rate (↑) |
|---------|----------------|----------------|---------|--------|------------------|
| Reference | -6.36 | -7.45 | 0.48 | 0.73 | 25.0% |
| AR | -5.75 | -6.75 | 0.51 | 0.63 | 7.1% |
| liGAN | - | -6.33 | 0.39 | 0.59 | 3.9% |
| GraphBP | - | -4.80 | 0.43 | 0.49 | 0.1% |
| Pocket2Mol | -5.14 | -7.15 | **0.56** | 0.74 | 24.4% |
| TargetDiff | -5.47 | -7.80 | 0.48 | 0.58 | 10.5% |
| DecompDiff | -5.67 | -8.39 | 0.45 | 0.61 | 24.5% |
| 3DMolFormer | **-6.02** | **-9.48** | 0.49 | **0.78** | **85.3%** |

**Results**  As shown in Table 2, the molecules designed by 3DMolFormer outperform those of all baselines across four metrics: Vina Score, Vina Dock, SA, and Success Rate. Notably, it exhibits a significant advantage in Success Rate, becoming the first method to exceed the reference values provided in the dataset for this key metric. Additionally, the result of QED also significantly surpasses the predefined threshold. This indicates that 3DMolFormer demonstrates superior performance in binding affinity optimization and multi-objective joint optimization compared to existing 3D drug design methods, highlighting its strong potential for real-world applications in drug discovery.

For further details, results, and a case study of experiments on pocket-aware 3D drug design, please refer to Appendix D.

## 5 CONCLUSION AND DISCUSSION

In this paper, we introduce 3DMolFormer for structure-based drug discovery, a dual-channel transformer-based framework designed to process parallel sequences of tokens and numerical values representing pocket-ligand complexes. Through self-supervised large-scale pre-training and supervised fine-tuning, 3DMolFormer can accurately and efficiently predict the binding poses of ligands to protein pockets. Furthermore, through reinforcement learning fine-tuning, 3DMolFormer can generate drug candidates that exhibit high binding affinities for a given protein target, along with favorable drug-likeness and synthesizability. Above all, 3DMolFormer is the first machine learning framework that can simultaneously address both protein-ligand docking and pocket-aware 3D drug design, and it outperforms previous baselines in both tasks.

It is noteworthy that many recent deep learning models for 3D molecules, such as Uni-Mol, Pocket2Mol, TargetDiff, and DecompDiff, which serve as baselines in our experiments, adhere to the concept of "equivariance" introduced by geometric deep learning (Atz et al., 2021; Isert et al., 2023a). However, the 3DMolFormer model does not explicitly enforce SE(3)-symmetry. It appears that through the normalization of 3D coordinates and random rotations during data augmentation, 3DMolFormer has acquired the SE(3)-equivariance by training on a sufficiently large and diverse dataset. This approach aligns with recent successful methods in the field, including AlphaFold3 (Abramson et al., 2024), which also does not rely on SE(3)-equivariant architectures.

Admittedly, our approach still has some limitations. First, 3DMolFormer does not account for the flexibility of proteins during ligand binding, which may affect the accuracy of subsequent binding affinity prediction. Second, protein-ligand binding is a dynamic process, but 3DMolFormer struggles to capture this dynamism effectively. Finally, 3DMolFormer does not consider environmental factors such as temperature and pH, which can significantly influence the 3D conformation of the binding complex. These issues represent core challenges in current computational methods for structure-based drug discovery, and we look forward to future work addressing these limitations. Furthermore, the implementation details in 3DMolFormer have the potential to be further optimized, for example, advanced methods of multi-objective reinforcement learning (Liu et al., 2014) may be introduced into the drug design process.

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

## A   PARALLEL SEQUENCES

Here is a real example of the parallel sequence of a pocket-ligand complex with a total length of 867, corresponding to Figure 1 and 2. The token sequence mainly consists of 4 parts: pocket atoms, pocket atom coordinates, ligand SMILES, ligand atom coordinates, and the start and end of each part are marked by special tokens. The first amino acid of the pocket is particularly marked. Moreover, the '[x]', '[y]', '[z]' tokens are corresponding to values representing 3D coordinates in the numerical sequence.

Tokens: ['[PS]', 'N', 'CA', 'C', 'O', 'C', 'C', 'C', 'N', 'CA', 'C', 'O', 'C', 'C', 'C', 'N', 'CA', 'C', 'O', 'C', 'C', 'N', 'C', 'C', 'N', 'N', 'CA', 'C', 'O', 'C', 'C', 'C', 'O', 'O', 'N', 'CA', 'C', 'O', 'C', 'C', 'C', 'N', 'CA', 'C', 'O', 'C', 'O', 'N', 'CA', 'C', 'O', 'C', 'O', 'C', 'N', 'CA', 'C', 'O', 'C', 'C', 'C', 'C', 'N', 'C', 'C', 'C', 'C', 'C', 'N', 'CA', 'C', 'O', 'C', 'C', 'N', 'C', 'C', 'N', 'N', 'CA', 'C', 'O', 'C', 'C', 'O', 'N', 'N', 'CA', 'C', 'O', 'C', 'O', 'C', 'N', 'CA', 'C', 'O', 'C', 'C', 'C', 'N', 'CA', 'C', 'O', 'C', 'C', 'C', 'N', 'CA', 'C', 'O', 'C', 'C', 'C', 'C', 'N', 'CA', 'C', 'O', 'C', 'C', 'C', 'C', 'N', 'CA', 'C', 'O', 'C', 'C', 'C', 'O', 'N', 'N', 'CA', 'C', 'O', 'C', 'C', 'N', 'C', 'C', 'N', 'N', 'CA', 'C', 'O', 'C', 'C', 'N', 'C', 'C', 'N', 'N', 'CA', 'C', 'O', 'C', 'C', 'O', 'N', 'N', 'CA', 'C', 'O', 'C', 'C', 'C', 'C', 'C', 'C', 'C', 'N', 'CA', 'C', 'O', 'C', 'C', 'C', 'C', 'N', 'CA', 'C', 'O', 'C', 'C', 'C', 'O', 'O', 'O', 'O', '[PE]', '[PCS]', '[x]', '[y]', '[z]', '[x]', '[y]', '[z]', '[x]', '[y]', '[z]', '[x]', '[y]', '[z]', '[x]', '[y]', '[z]', '[x]', '[y]', '[z]', '[x]', '[y]', '[z]', '[x]', '[y]', '[z]', '[x]', '[y]', '[z]', '[x]', '[y]', '[z]', '[x]', '[y]', '[z]', '[x]', '[y]', '[z]', '[x]', '[y]', '[z]', '[x]', '[y]', '[z]', '[x]', '[y]', '[z]', '[x]', '[y]', '[z]', '[x]', '[y]', '[z]', '[x]', '[y]', '[z]', '[x]', '[y]', '[z]', '[x]', '[y]', '[z]', '[x]', '[y]', '[z]', '[x]', '[y]', '[z]', '[x]', '[y]', '[z]', '[x]', '[y]', '[z]', '[x]', '[y]', '[z]', '[x]', '[y]', '[z]', '[x]', '[y]', '[z]', '[x]', '[y]', '[z]', '[x]', '[y]', '[z]', '[x]', '[y]', '[z]', '[x]', '[y]', '[z]', '[x]', '[y]', '[z]', '[x]', '[y]', '[z]', '[x]', '[y]', '[z]', '[x]', '[y]', '[z]', '[x]', '[y]', '[z]', '[x]', '[y]', '[z]', '[x]', '[y]', '[z]', '[x]', '[y]', '[z]', '[x]', '[y]', '[z]', '[x]', '[y]', '[z]', '[x]', '[y]', '[z]', '[x]', '[y]', '[z]', '[x]', '[y]', '[z]', '[x]', '[y]', '[z]', '[x]', '[y]', '[z]', '[x]', '[y]', '[z]', '[x]', '[y]', '[z]', '[x]', '[y]', '[z]', '[x]', '[y]', '[z]', '[x]', '[y]', '[z]', '[x]', '[y]', '[z]', '[x]', '[y]', '[z]', '[x]', '[y]', '[z]', '[x]', '[y]', '[z]', '[x]', '[y]', '[z]', '[x]', '[y]', '[z]', '[x]', '[y]', '[z]', '[x]', '[y]', '[z]', '[x]', '[y]', '[z]', '[x]', '[y]', '[z]', '[x]', '[y]', '[z]', '[x]', '[y]', '[z]', '[x]', '[y]', '[z]', '[x]', '[y]', '[z]', '[x]', '[y]', '[z]', '[x]', '[y]', '[z]', '[x]', '[y]', '[z]', '[x]', '[y]', '[z]', '[x]', '[y]', '[z]', '[x]', '[y]', '[z]', '[x]', '[y]', '[z]', '[x]', '[y]', '[z]', '[x]', '[y]', '[z]', '[x]', '[y]', '[z]', '[x]', '[y]', '[z]', '[x]', '[y]', '[z]', '[x]', '[y]', '[z]', '[x]', '[y]', '[z]', '[x]', '[y]', '[z]', '[x]', '[y]', '[z]', '[x]', '[y]', '[z]', '[x]', '[y]', '[z]', '[x]', '[y]', '[z]', '[x]', '[y]', '[z]', '[x]', '[y]', '[z]', '[x]', '[y]', '[z]', '[x]', '[y]', '[z]', '[x]', '[y]', '[z]', '[x]', '[y]', '[z]', '[x]', '[y]', '[z]', '[x]', '[y]', '[z]', '[x]', '[y]', '[z]', '[x]', '[y]', '[z]', '[x]', '[y]', '[z]', '[x]', '[y]', '[z]', '[x]', '[y]', '[z]', '[x]', '[y]', '[z]', '[x]', '[y]', '[z]', '[x]', '[y]', '[z]', '[x]', '[y]', '[z]', '[x]', '[y]', '[z]', '[x]', '[y]', '[z]', '[x]', '[y]', '[z]', '[x]', '[y]', '[z]', '[x]', '[y]', '[z]', '[x]', '[y]', '[z]', '[x]', '[y]', '[z]', '[x]', '[y]', '[z]', '[x]', '[y]', '[z]', '[x]', '[y]', '[z]', '[x]', '[y]', '[z]', '[x]', '[y]', '[z]', '[x]', '[y]', '[z]', '[x]', '[y]', '[z]', '[x]', '[y]', '[z]', '[x]', '[y]', '[z]', '[x]', '[y]', '[z]', '[x]', '[y]', '[z]', '[x]', '[y]', '[z]', '[x]', '[y]', '[z]', '[x]', '[y]', '[z]', '[x]', '[y]', '[z]', '[x]', '[y]', '[z]', '[PCE]', '[LS]', 'C', 'C', 'C', 'C', '(', 'C', '(', '=', 'O', ')', 'N', 'c', '1', 'c', 'c', 'c', '(', 'S', '(', 'N', ')', '(', '=', 'O', ')', '=', 'O', ')', 'c', 'c', '1', ')', 'C', '(', 'C', ')', '(', 'C', ')', 'C', '[LE]', '[LCS]', '[x]', '[y]', '[z]',

'[x]', '[y]', '[z]', '[x]', '[y]', '[z]', '[x]', '[y]', '[z]', '[x]', '[y]', '[z]', '[x]', '[y]', '[z]', '[x]', '[y]', '[z]', '[x]', '[y]', '[z]', '[x]', '[y]', '[z]', '[x]', '[y]', '[z]', '[x]', '[y]', '[z]', '[x]', '[y]', '[z]', '[x]', '[y]', '[z]', '[x]', '[y]', '[z]', '[x]', '[y]', '[z]', '[x]', '[y]', '[z]', '[x]', '[y]', '[z]', '[x]', '[y]', '[z]', '[x]', '[y]', '[z]', '[x]', '[y]', '[z]', '[x]', '[y]', '[z]', '[x]', '[y]', '[z]', '[x]', '[y]', '[z]', '[x]', '[y]', '[z]', '[x]', '[y]', '[z]', '[x]', '[y]', '[z]', '[x]', '[y]', '[z]', '[x]', '[y]', '[z]', '[x]', '[y]', '[z]', '[x]', '[y]', '[z]', '[LCE]'

Numbers: [ 1. 1. 1. 1. 1. 1. 1. 1. 1. 1. 1. 1. 1. 1. 1. 1. 1. 1. 1. 1. 1. 1. 1. 1. 1. 1. 1. 1. 1. 1. 1. 1. 1. 1. 1. 1. 1. 1. 1. 1. 1. 1. 1. 1. 1. 1. 1. 1. 1. 1. 1. 1. 1. 1. 1. 1. 1. 1. 1. 1. 1. 1. 1. 1. 1. 1. 1. 1. 1. 1. 1. 1. 1. 1. 1. 1. 1. 1. 1. 1. 1. 1. 1. 1. 1. 1. 1. 1. 1. 1. 1. 1. 1. 1. 1. 1. 1. 1. 1. 1. 1. 1. 1. 1. 1. 1. 1. 1. 1. 1. 1. 1. 1. 1. 1. 1. 1. 1. 1. 1. 1. 1. 1. 1. 1. 1. 1. 1. 1. 1. 1. 1. 1. 1. 1. 1. 1. 1. 1. 1. 1. 1. 1. 1. 1. 1. 1. 1. 1. 1. 1. 1. 1. 1. 1. 1. 1. 1. 1. 1. 1. 1. 1. 1. 1. 1. 1. 1. 1. 1. 1. 1. 1. 1. 1. 1. 1. 1. 1. 1. 1. 1. 1. 1. 1. 1. 1. 1. 1. 1. 1. 1. 1. 1. 1. 1. 1. 1. 1. 1. 1. 1. 1. 1. 1.258 -0.361 0.197 1.411 -0.114 0.217 1.242 0.118 0.107 1.106 0.083 -0.099 1.655 -0.181 0.045 1.693 -0.462 0.086 1.418 -0.577 0.076 -0.424 0.62 1.849 -0.673 0.475 1.784 -0.824 0.383 2.028 -0.714 0.248 2.201 -0.619 0.224 1.609 -0.881 0.084 1.542 -0.463 0.299 1.344 -1.458 -0.708 -0.818 -1.275 -0.894 -0.94 -1.285 -1.149 -0.778 -1.413 -1.159 -0.568 -0.986 -0.789 -0.949 -0.871 -0.739 -0.676 -0.726 -0.929 -0.54 -0.866 -0.515 -0.525 -0.637 -0.826 -0.313 -0.723 -0.575 -0.298 -0.515 -2.414 0.742 -0.313 -2.275 0.905 -0.378 -2.308 1.201 -0.227 -2.425 1.35 -0.31 -1.984 0.824 -0.231 -1.956 0.529 -0.09 -1.698 0.47 -0.209 -1.479 0.513 0.143 -1.706 0.383 -1.755 0.086 0.046 -1.589 0.29 0.172 -1.731 0.562 0.179 -1.814 0.664 -0.031 -1.318 0.318 0.019 -1.14 0.526 0.148 -1.175 0.045 -0. 0.011 -0.174 1.928 0.171 -0.047 1.719 0.083 -0.128 1.437 -0.074 -0.315 1.403 0.465 -0.129 1.757 0.514 -0.4 1.67 0.621 -0.628 0.417 0.846 -0.613 0.228 1.008 -0.363 0.288 0.909 -0.172 0.409 0.755 -0.591 -0.066 0.623 -0.345 -0.106 0.566 -0.814 -0.142 -1.305 -0.168 2.133 -1.388 -0.408 1.991 -1.66 -0.492 2.095 -1.703 -0.508 2.335 -1.183 -0.637 2.028 -0.97 -0.61 1.832 -0.729 -0.497 1.871 -0.981 -0.692 1.556 -0.588 -0.495 1.631 -0.736 -0.62 1.438 -1.174 -0.821 1.399 -0.679 -0.674 1.169 -1.117 -0.877 1.134 -0.873 -0.801 1.022 -0.812 -1.891 -0.518 -0.571 -2.055 -0.494 -0.657 -2.345 -0.518 -0.869 -2.419 -0.416 -0.439 -2.009 -0.222 -0.382 -1.72 -0.162 -0.133 -1.605 -0.2 -0.544 -1.519 -0.08 -0.143 -1.349 -0.135 -0.392 -1.289 -0.072 0.109 -0.508 -2.274 0.335 -0.345 -2.184 0.588 -0.505 -2.131 0.786 -0.399 -2.029 0.262 -0.179 -1.937 0.193 -0.35 -1.699 0.254 -0.585 -1.691 0.056 -0.232 -1.505 0.249 -0.548 0.808 0.382 -0.809 0.806 0.581 -0.839 0.578 0.694 -1.052 0.553 0.189 -1.044 0.78 0.07 -1.033 0.52 -0.029 -1.041 0.999 1.244 0.351 0.238 1.426 0.432 0.457 1.353 0.35 0.742 1.477 0.45 0.926 1.424 0.737 0.433 1.153 0.801 0.335 1.082 0.581 0.142 -0.177 1.888 0.87 0.004 1.664 0.936 0.224 1.742 1.135 0.36 1.565 1.242 0.132 1.522 0.692 -0.081 1.397 0.519 0.304 1.71 0.529 -1.008 1.286 1.446 -1.146 1.092 1.27 -1.317 0.904 1.433 -1.261 0.856 1.667 -0.944 0.926 1.111 -0.747 1.057 0.919 -0.636 0.849 0.724 -0.832 1.303 0.773 -1.73 0.636 -0.746 -1.678 0.408 -0.922 -1.728 0.154 -0.766 -1.731 0.16 -0.52 -1.389 0.408 -1.035 -1.167 0.434 -0.826 -1.025 0.176 -0.756 -1.137 -0.044 -0.769 -0.777 0.206 -0.67 -1.603 -1.141 -0.028 -1.559 -0.893 0.123 -1.694 -0.668 -0.013 -1.614 -0.591 -0.235 -1.259 -0.833 0.14 -1.116 -1.027 0.314 -0.843 -1.072 0.304 -1.214 -1.193 0.506 -0.78 -1.259 0.481 -1.002 -1.336 0.604 0.647 -1.02 -1.743 0.624 -1.167 -1.485 0.346 -1.28 -1.438 0.319 -1.491 -1.312 0.727 -1.026 -1.229 0.706 -0.729 -1.225 0.506 -0.597 -1.088 0.872 -0.537 -1.32 0.543 -0.337 -1.111 0.763 -0.295 -1.251 -0.888 -0.494 -1.826 -1.057 -0.313 -1.668 -1.3 -0.231 -1.836 -1.271 -0.142 -2.065 -0.913 -0.058 -1.581 -0.693 -0.102 -1.386 -0.672 -0.316 -1.267 -0.52 0.1 -1.351 -0.84 2.384 -0.05 -0.733 2.123 0.03 -0.443 2.139 0.11 -0.361 2.014 0.306 -0.766 1.927 -0.202 -0.663 1.652 -0.146 -0.809 1.472 0.01 -0.418 1.569 -0.244 -0.718 1.22 0.063 -0.324 1.316 -0.193 -0.478 1.138 -0.037 0.189 0.013 1.237 0.194 -0.096 0.964 0.344 -0.358 0.967 0.548 -0.396 1.111 0.343 0.095 0.776 0.232 0.378 0.764 0.407 0.536 0.569 -0.061 0.389 0.682 0.912 0.482 1.165 0.846 0.7 1.349 0.635 0.619 1.554 0.435 0.487 1.489 0.752 0.946 1.188 0.974 1.064 1.009 0.85 1.269 0.822 1.203 1.191 1.169 -1.592 0.135 1.225 -1.603 -0.071 1.018 -1.794 -0.288 1.113 -1.763 -0.394 1.334 -1.325 -0.19 0.955 -1.346 -0.396 0.728 -1.137 0.029 0.868 0.794 0.122 -0.903 -0.524 0.748 -0.693 0.269 -1.24 -0.704 0.718 0.231 -0.421 1. 1. 1. 1. 1. 1. 1. 1. 1. 1. 1. 1. 1. 1. 1. 1. 1. 1. 1. 1. 1. 1. 1. 1. 1. 1. 1. 1. 1. 1. 1. 1. 1. 1. 1. 1. 1. 1. 1. 1. 1. 1. 1. 1. 1. 1. 1. 1. 0.637 1.213 -0.028 0.463 0.997 -0.136 0.502 0.924 -0.423 0.258 0.857 -0.581 0.091 0.665 -0.42 -0.147 0.711 -0.403 0.202 0.434 -0.286 0.039 0.245 -0.136 0.171 0.024 -0.035 0.039 -0.171 0.113 -0.234 -0.159 0.155 -0.382 -0.425 0.342 -0.278 -0.713 0.219 -0.674 -0.386 0.319 -0.287 -0.384 0.617 -0.374 0.062 0.048 -0.238 0.262 -0.095 0.319 0.763 -0.862 0.596 0.657 -0.885 0.295 0.994 -1.051 0.123 0.552 -0.948 1. ]

In addition, the factor $q$ in Eq. (1) is set to 5.0. The performance of the model is not sensitive to the choice of $q$, because most of the 3D coordinates in our data (protein pockets and ligands) are in a limited range.

## B  PRE-TRAINING

**Data Source**   pockets for pre-training (3.2M), ligand conformations for pre-training (209M), and ground-truth protein-ligand complexes for docking fine-tuning (17K): `https://github.com/deepmodeling/Uni-Mol/tree/main/unimol`.

Docked protein-ligand complexes for pre-training and test set for pocket-aware 3D drug design: `https://github.com/guanjq/targetdiff`.

In addition, samples with the maximal difference in coordinates in one dimension greater than 40 are removed in order to filter out those outliers that account for less than 0.1% data.

**Model Scaling**   The standard dual-channel model used in our paper follows the configuration of the GPT-2 small model (Radford et al., 2019). An ablation study of pre-training is conducted to determine the appropriate scale for 3DMolFormer, where the pre-training loss on the ligand validation set is reported for each model size:

| Layers | Heads | Embedding length | Pre-training Loss |
|--------|-------|------------------|-------------------|
| 8      | 8     | 256              | 0.325             |
| 12     | 8     | 256              | 0.254             |
| 12     | 12    | 256              | 0.229             |
| **12** | **12**| **768**          | **0.178**         |
| 16     | 12    | 768              | 0.178             |
| 16     | 16    | 768              | 0.180             |

The standard model size achieves the best performance compared with others, as a result of which it is utilized in our design.

**Hyper-parameters**   The coefficient $\alpha$ in the loss function of Eq. (2) is set to 1.0. In an ablation study, we observe that the selection of $\alpha$ does not significantly affect the balance between CE loss and MSE loss:

| $\alpha$ | CE Loss | MSE Loss |
|----------|---------|----------|
| 0.1      | 0.164   | 0.014    |
| 1.0      | 0.164   | 0.014    |
| 10.0     | 0.164   | 0.015    |

This may be because the errors on the token sequences and those on the numerical sequences converge respectively during the large-scale pre-training.

The selection of other hyper-parameters follows the common practice of the pre-training of large language models (Radford et al., 2019; Quinn et al., 2019).

## C  PROTEIN-LIGAND BINDING POSE PREDICTION SUPPLEMENT

**Docking Setup**   The exhaustiveness of all 4 search-based docking baselines in Table 1 is set to 8, following the settings in Uni-Mol (Zhou et al., 2023a).

**Standard Errors**   As shown in Table 3, the standard errors of the 3DMolFormer performance results are obtained by 5 individual runs of supervised fine-tuning on protein-ligand docking. The minor standard errors further validate the robustness and soundness of 3DMolFormer.

Table 3: Standard Errors of 3DMolFormer performance results in Table 1.

| Methods | %<1.0Å (↑) | %<2.0Å (↑) | %<3.0Å (↑) | %<5.0Å (↑) | Avg. (↓) |
|---|---|---|---|---|---|
| 3DMolFormer | 43.8±0.4 | 84.9±0.5 | 96.4±0.2 | 98.8±0.0 | 1.29±0.02 |

**Additional Experiments on PoseBusters**    PoseBusters (Buttenschoen et al., 2024) s a widely-used benchmark for evaluating protein-ligand docking methods, particularly focusing on the challenges of blind docking, where the binding pocket information is not provided. However, in our study, we evaluate 3DMolFormer on PoseBusters using pocket information, providing a different evaluation context compared to typical PoseBusters assessments conducted for state-of-the-art docking approaches such as AlphaFold (Abramson et al., 2024), Chai-1 (Discovery et al., 2024), and Uni-Mol Docking V2 (Alcaide et al., 2024).

For experiments on PoseBusters, the blind docking baselines following the standard evaluation setup include AutoDock Vina (Trott & Olson, 2010), DiffDock (Corso et al., 2023), Uni-Mol Docking V2 (Alcaide et al., 2024), AlphaFold3 (Abramson et al., 2024), and Chai-1 (Discovery et al., 2024). For pocket-aware docking approaches including Uni-Mol (Zhou et al., 2023a) and our 3DMolFormer, we provide pocket information for docking. As shown in Table 4, our 3DMolFormer achieves a higher pocket-aware docking accuracy than Uni-Mol, which is also higher than the blind docking accuracy of all state-of-the-art baselines.

Table 4: Experimental results of protein-ligand binding pose prediction on PoseBusters benchmark.

| Methods | %<2.0Å (↑) |
|---|---|
| AutoDock Vina | 52.3 |
| DiffDock | 37.9 |
| Uni-Mol Docking V2 | 77.6 |
| AlphaFold3 | 76.3 |
| Chai-1 | 77.1 |
| Uni-Mol (pocket-aware) | 74.8 |
| 3DMolFormer (pocket-aware) | 81.5 |

## D  POCKET-AWARE 3D DRUG DESIGN SUPPLEMENT

**Clarification on the SA score**    It should be clarified that the SA score ranges in $[1, 10]$ as defined in the original paper (Ertl & Schuffenhauer, 2009), where a lower score is better. Following the previous work on pocket-aware 3D drug design (Guan et al., 2023b), we report the linearly transformed SA score: $SA = (10 - SA_{origin})/9 \in [0, 1]$, where a higher score is better.

**Clarification on molecular diversity**    We do not include metrics for molecular diversity such as internal diversity Benhenda (2018) and Hamiltonian diversity Hu et al. (2024) in out evaluation, because existing metrics are all based on 2D graph structures, while pocket-aware 3D drug design is a 3D molecular generation task.

**Generation Setup**    In addition, in the 3D drug design experiments no more than 100 molecules are generated by each baseline method for each protein pocket. For 3DMolFormer, exactly 100 unique molecules are generated and seleted for each protein pocket, which is a more stringent requirement.

**Case Study**    Visualization of the reference binding molecule and two molecules generated by 3DMolFormer on protein **4H3C**:

**Standard Errors and Ablation Study**    As shown in Table 5, the standard errors of the 3DMol-Former performance results are obtained by 5 individual runs of RL fine-tuning on pocket-aware 3D drug design. The minor standard errors further validate the robustness and soundness of 3DMol-Former.

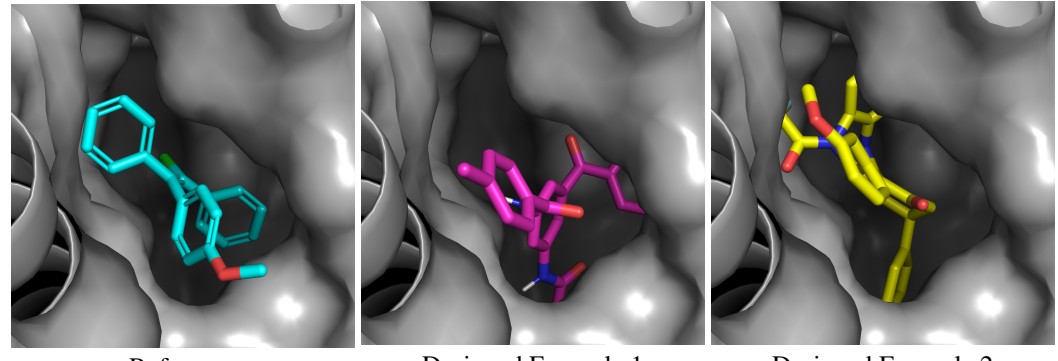

| Reference | Designed Example 1 | Designed Example 2 |

| Molecule | Vina Dock | QED | SA |
|----------|-----------|-----|-----|
| Reference | -8.0 | 0.55 | 0.91 |
| Designed Example 1 | -11.1 | 0.35 | 0.91 |
| Designed Example 2 | -10.6 | 0.48 | 0.75 |

In addition, we conduct an ablation study on 3DMolFormer. The variant 3DMolFormer w/o RL refers to freezing the GPT weights for RL fine-tuning, that is, generating molecules without fine-tuning for pocket-aware 3D drug design. The results indicate that the RL fine-tuning process is fundamental for this task.

Table 5: Standard Errors of 3DMolFormer performance results in Table 2, and results of the ablation study.

| Methods | Vina Score ($\downarrow$) | Vina Dock ($\downarrow$) | QED ($\uparrow$) | SA ($\uparrow$) | Success Rate ($\uparrow$) |
|---------|-----------|-----------|-----|-----|--------------|
| 3DMolFormer | -6.02±0.27 | -9.48±0.18 | 0.49±0.01 | 0.78±0.01 | 85.3%±1.5% |
| 3DMolFormer w/o RL | -4.20 | -5.03 | 0.46 | 0.50 | 2.1% |

**Distribution of Generated Molecules** Figure 4 demonstrates the distributions of molecular weights, logP values, and the number of rotatable bonds of the 10,000 molecules designed by 3DMolFormer for all the 100 targets reported in Table 2. It is worth mentioning that all three metrics are taken into account in drug-likeness, as measured by QED.

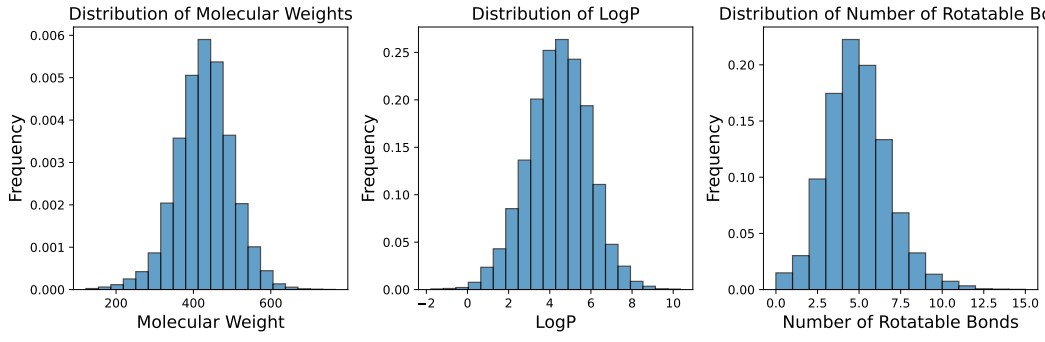

Figure 4: The distributions of molecular weights, logP values, and the number of rotatable bonds of the molecules designed by 3DMolFormer.

**Additional Evaluation by Delta Score** Delta Score is a novel evaluation metric in structure-based drug design that emphasizes the specificity of molecular binding (Gao et al., 2024b). Unlike

traditional docking scores that can inflate results due to biases, Delta Score evaluates the selective affinity of a molecule for its target compared to other potential binding pockets, providing a more accurate measure of binding specificity and reducing the influence of promiscuous binding effects.

Results in Table 6 show that 3DMolFormer outperforms previous methods in terms of Delta Score, demonstrating its superior capability to generate molecules with higher specificity for their intended targets.

Table 6: Experimental results of Delta Score on pocket-aware 3D drug design.

| Methods | Mean Delta Score (↑) |
|---|---|
| Reference | 1.158 |
| AR | 0.393 |
| Pocket2Mol | 0.437 |
| TargetDiff | 0.335 |
| DecompDiff | 0.354 |
| 3DMolFormer | **0.716** |

**Additional Evaluation by PoseCheck**   Clash Score and Strain Energy are key metrics used in PoseCheck (Harris et al., 2023) to evaluate the physical plausibility and stability of protein-ligand poses in structure-based drug design. Clash Score assesses steric clashes between atoms in the generated pose, while Strain Energy quantifies the energetic distortion from ideal molecular conformations. Both metrics ensure that generated poses align with physical and chemical principles.

Table 7 demonstrates that 3DMolFormer outperforms baselines on both metrics, highlighting its ability to produce more physically realistic and energetically favorable docking poses.

Table 7: Experimental results of Delta Score on pocket-aware 3D drug design.

| Methods | Mean Clash Score (↓) | Median Strain Energy (↓) |
|---|---|---|
| Reference | 4.59 | 102.5 |
| LiGAN | 3.40 | 18693.8 |
| Pocket2Mol | 5.62 | 194.9 |
| TargetDiff | 9.08 | 1241.7 |
| 3DMolFormer | **3.25** | **183.3** |

Furthermore, DrugPose (Jocys et al., 2024) offers a broad range of metrics for 3D drug discovery, its overlap with PoseCheck in the context of structure-based drug design makes PoseCheck a sufficient benchmark for our evaluation, ensuring comprehensive assessment without redundancy.

