# OpenReview forum: "3DMolFormer: A Dual-channel Framework for Structure-based Drug Discovery"
_ICLR.cc/2025/Conference — ICLR 2025 Poster_

### Official Review · Reviewer_H1Mo · 2024-11-02

**Soundness:** 2
**Presentation:** 2
**Contribution:** 2
**Rating:** 5
**Confidence:** 4

**Summary:**

The paper presents 3DMolFormer, a dual-channel transformer framework designed for protein-ligand docking and pocket-aware 3D drug design. It utilizes a parallel sequence format to represent 3D pocket-ligand complexes and employs a "pre-training + fine-tuning" approach on a large dataset to model 3D information effectively.

**Strengths:**

Unified Framework: 3DMolFormer integrates protein-ligand docking and pocket-aware 3D drug design into a single model.
Parallel Sequence Format: Represents 3D pocket-ligand complexes, facilitating effective modeling of both discrete and continuous information.
Large-Scale Pre-training: Utilizes a pre-training.
Enhanced 3D Information Modeling: Effectively addresses challenges in modeling complex 3D interactions.
Novel dual generative+docking : Applicable to multiple tasks within structure-based drug discovery with 3D information, improving efficiency in drug design processes.

**Weaknesses:**

Docking. The model should be benchmarked against PoseBusters. The state of the art here includes AF3 and Chai-1, which achieve an accuracy of 76–77%. These models are only trained on PDB data. I understand that cofolding is not part of this approach, but we still need to assess the trade-offs on a widely recognized benchmark.

Synthetic Accessibility Score. It's stated that a score above 0.59 is given a value of 1. Why is this the case? Typically, a lower score indicates better synthetic accessibility. Is there a mistake, or are they using a different metric? I noticed that further clarification is provided in the Appendix.

Multi-score Thresholds. The thresholds for multi-score criteria appear to be chosen arbitrarily. For example, a value of 1 is assigned to QED and SA scores that surpass a certain threshold. Could you clarify how these cutoffs were selected?

Docking Formula. They provide a formula for redocking, but typically, a different target would require a distinct reference score, standardized by weight. There doesn’t seem to be a standard docking energy applied here.

Molecule Distribution. The paper does not specify the distribution of generated molecules in terms of size. Is there variation in logP values? How about the number of rotatable bonds?

Benchmarks for 3D Generation. I suggest using established benchmarks, such as CheckPose or DrugPose, for 3D generation.

**Questions:**

-Could you clarify how well you perform on PoseBusters?
-What are speed accuracy trade-offs in comparison to AlphaFold3 or Chai-1?
-How the thresholds where chosen for the addition. By calculating using the formula. It seems that it was chosen to use QED 4.39?
-What are the proteins in which you generated the molecules? How did you choose them? Are they diverse?

---

> ### Author Response · Authors · 2024-11-20
> **Responses to Reviewer H1Mo**
>
> We greatly appreciate your thorough review and valuable feedback. Below, we provide detailed responses to each of your concerns and hope this will lead you to reconsider your evaluation.
>
> ## Response to Weaknesses
>
> 1. **Docking**:
>    - Thank you for suggesting the inclusion of **PoseBusters** as a benchmark. We have conducted experiments and included results in **Appendix C** of our revised paper. However, it is important to note that PoseBusters is designed primarily for blind docking, whereas our 3DMolFormer focuses on pocket-aware docking. To bridge this difference, we evaluated 3DMolFormer on PoseBusters with provided pocket information, representing a different setup from typical PoseBusters evaluations. We have discussed this further in **Section 4.1**.
>    - **Speed and Accuracy Trade-offs**: While 3DMolFormer excels in scalability due to parallel inference, direct speed-accuracy comparisons with models like AlphaFold3 and Chai-1 are not meaningful, as they target different tasks. Only for context, 3DMolFormer predicts a binding pose in an average of 0.8 seconds using an A100 80G GPU, whereas AF3 takes over 60 seconds. We have argued this in **Section 4.2 - Results**.
> 2. **Synthetic Accessibility (SA) Score**:
>    - The original SA score ranges from [1, 10], where lower values indicate better synthetic accessibility. For consistency with previous studies such as Pocket2Mol and DecompDiff, we applied a negative linear transformation to scale the SA score to [0, 1], where higher values are preferable. The threshold of 0.59 follows DecompDiff, where scores above this threshold signify good synthetic accessibility. We have clarified this in **Section 4.2 - Footnote** and **Appendix D**.
> 3. **Multi-Score Thresholds**:
>    - The thresholds for Vina Dock < −8.18, QED > 0.25, and SA > 0.59 were derived from DecompDiff to ensure comparability with established baselines. Moreover, QED and SA scores above their thresholds are assigned a reward value of 1 since they are auxiliary to the main docking score. We have included further clarification in **Section 4.2 - Reward Function** of the revised manuscript.
> 4. **Docking Formula**:
>    - You correctly noted that we did not use a fixed threshold for the Vina Dock score in our reward function. Instead, we aimed to minimize this score as much as possible. We used a reverse sigmoid function, as discussed in **Section 4.2 - Reward Function**, to convert the Vina Dock score into a [0, 1] range where higher values are preferred.
> 5. **Molecule Distribution**:
>    - We appreciate your suggestion regarding molecule distribution. We now include the distributions of molecular weights, logP values, and the number of rotatable bonds in **Appendix D**. Although these distributions are not used for direct comparison, their contributions are captured within the QED metric, which reflects drug-likeness.
> 6. **Benchmarks for 3D Generation**:
>    - We appreciate your recommendation to include benchmarks such as PoseCheck and DrugPose. We have incorporated **clash scores** and **strain energy** assessments from PoseCheck to evaluate the structural rationality of generated molecules. Results are included in **Appendix D**, which further validate the advantages of our 3DMolFormer.
>    - For DrugPose, while it offers various metrics for 3D drug discovery, its overlap with PoseCheck in the context of structure-based drug design makes PoseCheck a sufficient benchmark for our evaluation. We have included a discussion on this in **Appendix D**.
>
> ## Response to Questions
>
> - **PoseBusters Performance**: We present our new evaluation on PoseBusters in **Appendix C**, highlighting the difference in application context between blind and pocket-aware docking. (Also refer to Response to Weaknesses 1)
> - **Speed and Accuracy Comparisons**: We provide speed and performance insights in **Section 4.1**, noting that 3DMolFormer is highly scalable and efficient. However, due to different scope of tasks, the comparison of speed-accuracy trade-off between 3DMolFormer and AF3 or Chai-1 is meaningless. (Also refer to Response to Weaknesses 1)
> - **Threshold Selection Explanation**: The selection of thresholds—QED > 0.25, SA > 0.59, and Vina Dock < −8.18—follows standard benchmarks from DecompDiff for consistency, detailed in **Section 4.2**. (Also refer to Response to Weaknesses 3)
> - **Protein Selection**: As mentioned in **Section 4.2 - Data**, for the protein pockets for 3D drug design , we selected 100 diverse pockets from the CrossDocked2020 dataset, ensuring they had <30% sequence similarity to pre-training data to evaluate the generalizability of our model.
>
> ------
>
> We thank you again for your comprehensive review. We hope these clarifications address your concerns of our work.

---

> > ### Author Response · Authors · 2024-11-27
> > **Further responses to Reviewer H1Mo**
> >
> > Dear Reviewer H1Mo,
> >
> > We have responded to your concerns point-by-point in the rebuttal and updated the revised contents in the paper PDF for your reference. As the rebuttal deadline is approaching, we kindly request your feedback and ask if you have any additional questions. If our responses have addressed your points satisfactorily, we would appreciate it if you could update your score for our paper accordingly.
> >
> > Thank you for your time and consideration, and we look forward to hearing from you.
> >
> > Best regards,
> >
> > Authors

---

> > > ### Author Response · Authors · 2024-12-04
> > > **Thanks to Reviewer H1Mo**
> > >
> > > Dear Reviewer H1Mo,
> > >
> > > Thank you for your thoughtful feedback and for raising the score of our paper. We sincerely appreciate your recognition of the efforts we put into addressing your concerns.
> > >
> > > Best regards,
> > >
> > > Authors

---

### Official Review · Reviewer_HfQS · 2024-11-02

**Soundness:** 3
**Presentation:** 3
**Contribution:** 2
**Rating:** 6
**Confidence:** 3

**Summary:**

This paper proposes a novel dual-channel transformer-based framework that effectively handles both protein-ligand docking and structure-based drug design (SBDD) tasks.

**Strengths:**

1. Docking and structure-based drug design (SBDD) are indeed dual tasks. The method presented in this paper, which models both tasks simultaneously within a single framework, represents a promising and logical approach.
2. By leveraging the similar architecture of GPT, the proposed method demonstrates significant scalability, including both model parameters and data volume, allowing for the effective utilization of large-scale datasets for pre-training.

**Weaknesses:**

As discussed in Section 5, the proposed method does not consider SE(3) symmetry explicitly but instead relies on data augmentation techniques. I think this aspect warrants further discussion and consideration. Although the experiments validate the method's effectiveness to some extent, I believe the persuasive power of these findings is limited when considering the following points:
1. For docking task, as far as I know, the more advanced approach Uimol-docking v2 is not included in the baslines.
2. For sbdd task, I noticed that the article directly employs evaluation metrics (Vina QED SA) as the reward function and utilizes reinforcement learning for fine-tuning. This approach may be considered unfair to other methods.  It may be necessary to incorporate abalation study as well as additional evaluation methods to comprehensively validate the effectiveness of the approach, for example, the delta score proposed in paper [1] may serve as a metric for assessing whether the method is overfitting to the Vina evaluation.


[1]. Ren, Minsi, et al. "Delta Score: Improving the Binding Assessment of Structure-Based Drug Design Methods." arXiv preprint arXiv:2311.12035 (2023).

**Questions:**

Refer to the weaknesses.

**Details Of Ethics Concerns:**

NA.

---

> ### Author Response · Authors · 2024-11-20
> **Responses to Reviewer HfQS**
>
> Thank you for your valuable feedback on our paper. We appreciate your insightful comments and have revised the manuscript accordingly. Below, we provide our responses to your concerns point-by-point:
>
> > As discussed in Section 5, the proposed method does not consider SE(3) symmetry explicitly but instead relies on data augmentation techniques. I think this aspect warrants further discussion and consideration.
>
> We appreciate your attention to this aspect. While our method does not explicitly enforce SE(3) symmetry, it achieves implicit equivariance through data normalization and augmentation, such as random rotations, which provide sufficient coverage of the rotational and translational transformations. This approach aligns with recent successful methods in the field, including **AlphaFold3**, which also does not rely on explicit SE(3) equivariance. We have expanded the discussion in **Section 5** to highlight this rationale.
>
> > For docking task, as far as I know, the more advanced approach Unimol-docking v2 is not included in the baselines.
>
> Thank you for this suggestion. We have added **Unimol-docking v2** as a baseline in our docking task experiments on the **PoseBuster dataset**, aligning with the setup in the Unimol-docking v2 paper. These results can be found in **Appendix C** of the revised paper.
>
> It is important to note, however, that Unimol-docking v2 is primarily designed for **blind docking**, while our method, 3DMolFormer, is aimed at **pocket-aware docking**. We have clarified this scope difference in **Section 4.1** of the revised paper.
>
> > For sbdd task, I noticed that the article directly employs evaluation metrics (Vina QED SA) as the reward function and utilizes reinforcement learning for fine-tuning. This approach may be considered unfair to other methods. It may be necessary to incorporate ablation study as well as additional evaluation methods to comprehensively validate the effectiveness of the approach, for example, the delta score proposed in paper [1] may serve as a metric for assessing whether the method is overfitting to the Vina evaluation.
>
> You are correct in noting that we directly use the evaluation metrics to formulate the RL reward function. This reflects the **flexibility** of our framework, as it allows for targeted optimization towards specific objectives (i.e.., desirable molecular properties). In contrast, prior methods often only learn these properties implicitly from their training data.
>
> We agree that using the **delta score** as proposed in [1, 2] is a meaningful way to assess specificity in SBDD and check for potential overfitting. We have incorporated the **delta score** in our evaluation, and the results are detailed in **Appendix D**, which further reflect the advantages of our framework in SBDD.
>
> ------
>
> We hope these updates address your concerns and contribute to a clearer understanding of our work.

---

> ### Comment · Reviewer_HfQS · 2024-11-22
>
> 1. Concerns about the SE(3):
>
>  I also notice that AlphaFold3 does not employ SE(3)-equivariant architectures. In your opinion, is it possible that, in cases where the dataset is sufficiently large, simple data augmentation alone is adequate to address the SE(3)-equivariant properties? Additionally, what do the authors think about the potential pros and cons of using SE(3)-equivariant architectures compared to architectures that do not consider such properties?
>
> 2. Concerns about the docking:
>
> Thanks for the clarification. However, I also noticed that the results for the compared methods, such as AlphaFold3 and UniMol DockingV2, were obtained under the blind docking setup. Would providing more accurate pocket location information for these methods result in better performance?
>
> 3. Concerns about the SBDD:
>
> Thank you for the additional experiments, however, I have another concern. As I understand it, one of the key innovations of the paper is the use of a single model to unify the docking and SBDD tasks. However, from the ablation study in Appendix D, Table 5, I observed that the method seems to be highly dependent on reinforcement learning. In this case, what do the authors consider to be the benefit of unifying these two tasks for the SBDD task specifically? Or, could there be some RL-based methods that could be used for comparison?
>
> Thank you to the authors for their efforts. In general, I am open to increasing the score after receiving additional clarifications.

---

> ### Author Response · Authors · 2024-11-23
> **Further Response to Reviewer HfQS**
>
> Thank you for your thoughtful follow-up questions and additional concerns. We are grateful for the opportunity to provide further clarifications and have included our detailed responses below.
>
>
> 1. **Concerns about SE(3):**
>
>    This is indeed a meaningful and interesting question. SE(3)-equivariance can be viewed as a form of "domain knowledge." Incorporating this property into model design reduces the need for the model to learn these equivariances directly from the data, which is especially beneficial when the training dataset is small or lacks diversity. However, when datasets are sufficiently large, SE(3)-equivariance can also be learned through standard architectures with data augmentation techniques like random rotation, which intuitively injects SE(3)-equivariant properties into the training data.
>
>    Notably, AlphaFold3 [1] highlights that simplifying the architecture of AlphaFold2 by removing SE(3)-equivariant processing has only a modest impact on accuracy. This also leads to the "relatively standard architecture" of AlphaFold3. A recent study [2] draws the similar conclusion that "equivariance improves data efficiency, but training non-equivariant models with data augmentation can close this gap given sufficient epochs." Although this study focuses on rigid-body interactions, the findings suggest that SE(3)-equivariance can be compensated for by large datasets and longer training times.
>
>    In summary, SE(3)-equivariant architectures have the advantage of explicitly ensuring equivariance, particularly in data-limited scenarios, but come at the cost of increased architectural complexity. In contrast, simpler architectures paired with data augmentation offer scalability and efficiency, albeit with a higher dependency on large and diverse datasets. We appreciate this thought-provoking discussion and have included these points in **Section 5** of the revised paper.
>
> 2. **Concerns about the docking:**
>
>    Thank you for this question! Blind docking and pocket-aware docking are fundamentally different tasks with distinct inputs and training data, making direct comparisons difficult. Blind docking methods cannot leverage pocket information directly, as they are not trained or designed to incorporate it.
>
>    Specifically:
>
>    - **AlphaFold3**, which was recently open-sourced, does not include a channel for binding pocket input, making it infeasible to provide pocket information.
>    - **Uni-Mol Docking V2** is a variant of Uni-Mol designed explicitly for blind docking, transferring knowledge from pocket-aware docking (by using the weights of Uni-Mol) but trained for the blind docking task. Meanwhile, the original **Uni-Mol** has been included as a baseline for pocket-aware docking in **Section 4.1**.
>
> 3. **Concerns about SBDD:**
>
>    As we state in the abstract of the paper, our approach leverages the duality between docking and structure-based drug design (SBDD) by "utilizing docking functionalities within the drug design process." Specifically, the reinforcement learning (RL) process in 3DMolFormer is applied solely to the generation of ligand SMILES, while the protein-ligand docking ability, obtained from supervised fine-tuning, remains frozen during RL fine-tuning. This integration demonstrates the synergy of unifying these two tasks.
>
>    Regarding your observation on the dependency of the method on RL, this reflects the importance of RL in optimizing ligand generation within the unified framework. Before 3DMolFormer, structure-based 3D drug design was dominated by diffusion-based methods, where RL was not applicable. Our method introduces a distinct paradigm by employing a "pre-training + fine-tuning" approach tailored for transformer models, with RL fine-tuning being a widely adopted strategy in this context.
>
>    While RL-based methods for comparison are scarce, we acknowledge DeepLigBuilder [3], which employs Monte Carlo tree search (MCTS) for SBDD. However, the RL technique of MCTS differs significantly from our approach. Moreover, it designs drugs for only one protein target, whereas our model is evaluated across 100 targets. And DeepLigBuilder is not open-sourced, making it unavailable for direct comparison in our experiments.
>
> **Reference:**
>
> [1] Abramson J, Adler J, Dunger J, et al. Accurate structure prediction of biomolecular interactions with AlphaFold 3. *Nature*, 2024: 1–3.
>
> [2] Brehmer J, Behrends S, de Haan P, et al. Does equivariance matter at scale? *arXiv preprint arXiv:2410.23179*, 2024.
>
> [3] Li Y, Pei J, Lai L. Structure-based de novo drug design using 3D deep generative models. *Chemical Science*, 2021, 12(41): 13664–13675.
>
> ------
>
> Thank you again for your insightful questions. We hope these clarifications address your concerns and further highlight the contributions and strengths of our work. We greatly appreciate your openness to revisiting your evaluation and look forward to any additional feedback.

---

> ### Comment · Reviewer_HfQS · 2024-11-24
>
> Thank you to the authors for the detailed responses and discussions, which have largely resolved my concerns. My only remaining doubt is whether it is fair to directly use evaluation metrics (such as VINA) as the optimization objective for RL, since there have been previous approaches based on optimization theory that achieve better results than data-driven methods (such as diffusion-based methods), at least on the surface. However, this controversy remains an open question, and I will leave it to the AC to make the final decision. Additionally, I have raised the score to 6 for their efforts and improvements.

---

> > ### Author Response · Authors · 2024-11-24
> > **Further Response to Reviewer HfQS**
> >
> > We sincerely thank you for your prompt feedback and for raising the score! We deeply appreciate your acknowledgment of our efforts and the improvements made in response to your concerns.
> >
> > Regarding your remaining doubt, we believe reinforcement learning (RL) is a powerful and flexible technique for directly optimizing generative models toward specific objectives. RL has been widely adopted recently, especially in training large language models, to align their outputs with well-defined metrics or preferences. In the context of structure-based drug design, RL provides a direct pathway to optimize molecular properties, making it a suitable choice for our framework.
> >
> > As you rightly pointed out, diffusion-based and transformer-based architectures are two of the most popular generative paradigms in recent years. However, transformer architectures have not yet received as much attention in structure-based drug design as diffusion models. Through 3DMolFormer, we highlight the potential of transformer-based architectures in this field. We hope our work will inspire future research to further explore and refine these approaches, especially in combining RL with generative models for molecular design.
> >
> > Thank you once again for your insightful questions, constructive feedback, and positive responses. We truly appreciate your thoughtful engagement with our work.

---

### Official Review · Reviewer_69DH · 2024-11-02

**Soundness:** 3
**Presentation:** 2
**Contribution:** 3
**Rating:** 6
**Confidence:** 3

**Summary:**

This paper introduce 3DMolFormer, a dual-channel Transformer-based model that can process atom sequence and coordinate information parallelly, and thus this model is claimed to be the first one that could simultaneously address both protein-ligand docking and pocket-aware 3D drug design, and it outperforms previous baselines in both tasks.

**Strengths:**

1. This paper introduce a novel transformer-based model that can handle docking and structure-based drug design simultaneously

**Weaknesses:**

1. This paper mention figure 1 multiple times when introducing model structure, however there is no figure 1 in the preprint.

**Questions:**

1. For the pose evaluation, the model only check for RMSD. Could you also report other pose-relate metrics like steric clashes and strain-energy?

---

> ### Author Response · Authors · 2024-11-20
> **Responses to Reviewer 69DH**
>
> We appreciate your feedback and have addressed all of your concerns thoroughly, and we sincerely hope that you will reconsider your evaluation on our paper:
>
> > This paper mention figure 1 multiple times when introducing model structure, however there is no figure 1 in the preprint.
>
> **Figure 1 is located at the top of page 3, just above Figure 2.** To enhance clarity, we have highlighted this in the revised version of the manuscript to make it more prominent and easy to find.
>
> > For the pose evaluation, the model only check for RMSD. Could you also report other pose-relate metrics like steric clashes and strain-energy?
>
> Thank you for this insightful suggestion. We reviewed the methodology in *PoseCheck* [1], which introduced **clash scores** and **strain energy** as metrics for assessing the rationality of structure-based drug design, rather than protein-ligand docking. We have integrated these metrics into our evaluation and reported the results in **Appendix D** of the revised manuscript, which further confirm the advantages of our 3DMolFormer.
>
> [1] Harris, Charles, et al. "Posecheck: Generative models for 3d structure-based drug design produce unrealistic poses." *NeurIPS 2023 Generative AI and Biology (GenBio) Workshop*. 2023.

---

> > ### Comment · Reviewer_69DH · 2024-11-26
> >
> > Dear Authors,
> >
> > Thanks for addressing my concerns. My rating has raised to 6

---

> > > ### Author Response · Authors · 2024-11-27
> > >
> > > Dear Reviewer 69DH,
> > >
> > > We sincerely thank you for your feedback and for raising the score! We deeply appreciate your acknowledgment of our efforts and the improvements made in response to your concerns.
> > >
> > > Authors

---

### Official Review · Reviewer_U5kn · 2024-11-02

**Soundness:** 3
**Presentation:** 3
**Contribution:** 3
**Rating:** 8
**Confidence:** 4

**Summary:**

This paper introduces 3DMolFormer, a unified transformer-based framework for SBDD, which is capable of docking prediction and pocket-aware 3D drug design. The motivations of 3DMolFormer are clearly stated: current computational docking methods lack accuracy, while current 3D pocket aware drug design methods are unable to take full advantage of 3D structural information due to factors such as difficulties in 3D information modelling and limited data regarding ground-truth protein-ligand complexes. To represent a 3D complex of a protein pocket and a small molecule ligand, 3DMolFormer uses a parallel sequence composed of the SMILES atom token sequences for the protein and small molecule, along with the numerical sequence for 3D coordinates. A GPT architecture is pretrained through autoregressive generation of the parallel sequence. Fine-tuning consists of: (1) a supervised protein-ligand binding pose prediction task, and (2) a multi-objective RL pocket-aware molecular generation task. The presented results suggest that 3DMolFormer is successful in both fine-tuning tasks; it accurately predicts binding poses of ligands to protein pockets, and is capable of generating molecules that display high binding affinity to protein targets, while being synthesizable and exhibiting drug-like qualities.

**Strengths:**

- The proposed GPT framework seems interesting and presents novelty in terms of representing 3D complexes.
- 3DMolFormer presents strong results compared to other models on both fine-tuning tasks.
- The presentation of this paper is clear and well-structured.

**Weaknesses:**

- The multi-objective optimization of the RL seems to be overly simplistic, including a reward function that assigns a constraint-based reward for QED and SA. Literature on multi-obj DRL shows using more sophisticated reward functions and multi-objective optimization techniques greatly improve agent performance and stability.
- Minimal ablation studies are conducted, and all results are based on one run. More runs should be conducted to demonstrate the soundness of the model.

Minor edits:
- Line 482: the parameter σ in Eq. (4) was is to 100

**Questions:**

- The authors state: “... the sampling of ligand SMILES utilizes the weights of the RL agent’s model, which are continuously updated during finetuning. In contrast, the generation of atomic 3D coordinates uses the weights from the model finetuned for docking, which remains unchanged during this process.” Why don’t the authors freeze the GPT weights during the sampling of ligands? If this hinders the model’s performance, why is this not shown in the ablation studies?

---

> ### Author Response · Authors · 2024-11-20
> **Responses to Reviewer U5kn**
>
> We sincerely appreciate your positive feedback on the novelty, clarity, and strong performance demonstrated by 3DMolFormer. Below, we address your concerns point-by-point.
>
> ## Response to Weaknesses
>
> > The multi-objective optimization of the RL seems to be overly simplistic, including a reward function that assigns a constraint-based reward for QED and SA. Literature on multi-obj DRL shows using more sophisticated reward functions and multi-objective optimization techniques greatly improve agent performance and stability.
>
> Thanks for your suggestion! We understand and acknowledge that the reinforcement learning (RL) reward function in our work could be seen as simplistic compared to more sophisticated multi-objective optimization approaches in the literature of general machine learning. However, our design was intentionally straightforward to maintain focus on demonstrating the core innovations of our model, namely the dual-channel transformer framework and its applicability to both docking and drug design tasks. We have included an expanded discussion in **Section 5** of our revised paper to provide context for this design decision and outline potential extensions using advanced RL techniques.
>
> > Minimal ablation studies are conducted, and all results are based on one run. More runs should be conducted to demonstrate the soundness of the model.
>
> Thank you for this suggestion. In response, we have taken two steps to strengthen our experimental evidence:
>
> 1. **Reproducibility Across Runs**: We conducted additional experiments, running 3DMolFormer five times on both protein-ligand docking and pocket-aware 3D drug design tasks. We now report the standard error of these results in **Appendices C and D** of the revised paper, demonstrating the robustness and consistency of our model's performance across multiple runs.
> 2. **Ablation Study on RL Fine-Tuning**: We performed an ablation study to analyze the impact of freezing the weights for ligand SMILES generation during RL fine-tuning. The results, included in **Appendix D**, show that freezing these weights significantly reduces the quality of generated ligands and hinders reward optimization. This validates the importance of allowing weight updates during RL fine-tuning, as it enables the generation of molecules with increasingly higher expected rewards.
>
> >  **Minor Edits**: Line 482: the parameter σ in Eq. (4) was is to 100
>
> The typo in line 482 has been corrected in the revised paper.
>
> ## Response to Question
>
> > The authors state: “... the sampling of ligand SMILES utilizes the weights of the RL agent’s model, which are continuously updated during finetuning. In contrast, the generation of atomic 3D coordinates uses the weights from the model finetuned for docking, which remains unchanged during this process.” Why don’t the authors freeze the GPT weights during the sampling of ligands? If this hinders the model’s performance, why is this not shown in the ablation studies?
>
> Thank you for this insightful question. The core purpose of RL fine-tuning in our framework is to iteratively optimize the weights for ligand SMILES generation, ensuring that the generated molecules achieve progressively higher expected rewards. Freezing these weights would undermine the RL process, preventing any improvement in the ligand generation mechanism. Conversely, the generation of atomic 3D coordinates is not the target of RL fine-tuning, which is why we use frozen weights for this part.
>
> To substantiate this, we conducted an **ablation study** comparing performance with and without freezing the weights for ligand SMILES generation during RL fine-tuning. The results, detailed in **Appendix D**, reveal that freezing the weights significantly reduces reward optimization and degrades the quality of generated ligands. These findings underscore the necessity of weight updates in achieving the goals of RL fine-tuning and validate our design decisions.
>
> ------
>
> Thank you again for your valuable feedback. We hope our responses and revisions address your concerns adequately and further highlight the strengths of our work.

---

> > ### Comment · Reviewer_U5kn · 2024-11-20
> > **Response to authors**
> >
> > Dear Authors,
> >
> > Thanks for addressing my concerns. My rating will remain at 8, my confidence will increase.

---

> > > ### Author Response · Authors · 2024-11-20
> > >
> > > Dear Reviewer U5kn,
> > >
> > > Thank you for your thoughtful feedback and for taking the time to consider our rebuttal. We are grateful for your recognition of our work and are encouraged by your increased confidence in our submission.
> > >
> > > We appreciate the effort all reviewers have invested in providing detailed feedback, which has been invaluable in refining our work. We remain committed to ensuring clarity and addressing any concerns as the review process progresses.
> > >
> > > Thank you again for your valuable time and effort!
> > >
> > > Best regards,
> > >
> > > Authors

---

### Author Response · Authors · 2024-11-20
**Global Response**

We sincerely thank all reviewers for your valuable feedback and constructive suggestions. We have provided detailed point-by-point responses to each review to address your concerns.

In addition, we have uploaded a revised version of the paper. In this revised version, we use a **yellow background** to highlight the contents referenced in our responses and a **blue background** to indicate newly added content. These revisions are primarily located in **Sections 4 and 5**, as well as **Appendices C and D**. The updates include additional experiments, expanded discussions, and clarifications that align with the feedback provided.

We greatly appreciate your time and effort in reviewing our work and look forward to any further feedback you may have.

---

### Public Comment · ~Xiangyu_Huang3 · 2024-12-06
**Ask for more results**

Dear Authors,

I am working on 3D molecular generation, and I came across this paper and was very interested in the idea of framing 3D structure generation as language modeling, as in ESM3 there are very complicated tokenization designs, while in this paper the authors only input every Euclidean coordinate. However, this paper claims to *effectively addressing the challenges of modeling 3D information in SBDD*, yet with limited evaluation in terms of 3D structures, and I found current discussion inadequate for me to truly appreciate this paper’s value. In hopes for a more comprehensive evaluation in order to better understand the value of this paper and possibly help the authors to demonstrate their results, I would like to raise several questions about the method.

## More 3D metrics
It's very impressive to see the generated molecules with such a low strain energy and high affinities, and a more thorough demonstration would help to provide more insight.
- I am very curious about the Posebusters validity (PB-Valid) ratio for docking methods (since DL methods are known to violate the 3D physical constraints often) like in AlphaFold3, Chai-1, Boltz-1 and other recent docking methods. How does 3DMolFormer perform in terms of PB-Valid?
- Additionally, I checked the DecompDiff paper and found it important to assess bond length, bond angle and torsion angle distributions of the generated molecular structures. Can the authors provide the Jensen-Shannon Divergence between the molecules generated by their 3DMolFormer and reference molecules, too?

## Concerns on fair comparison
My biggest impression is that RL boosts the model’s performance so much, from barely working (2.1% Success Rate) to an astonishing level (85.3%). Can the authors explain why RL is so crucial here, and provide more implementation details? For example, on what dataset is the reward model being trained? What architecture is the reward model?

Moreover, as Reviewer HfQS pointed out, this paper might have made an unfair comparison between their approach (w/ RL) with non-optimization baselines (w/o RL). This brings out my third question.


## More literature review & SOTA baselines
I am not an expert in docking or drug design. But after checking the reference, it seemed to me that all the baselines are published no later than 2023, suggesting they are not SOTA now. Can the authors provide a more thorough list of related works and include the most recent models for comparison? Also, I am eager to see the missing literature review of molecule optimization, and how 3DMolFormer (w/ RL) compares with them.

Thank you very much! The authors’ feedback will be very helpful for my research.

---

> ### Public Comment · ~Xiangyu_Huang4 · 2024-12-06
>
> I'm Xiangyu Huang from Department of Life Science, Tsinghua University. I didn't write the comment above and I doubt that the commenter stole my personal information.

---

### Public Comment · ~Nahyun_Kim3 · 2025-08-10
**Questions regarding "3DMolFormer" and code reproducibility**

Dear Authors,

I recently read your paper "3DMolFormer: A Dual-channel Framework for Structure-based Drug Discovery" with great interest and was particularly impressed by the results. I would be very grateful for your guidance on a few points.


## 1. Generation fine-tuning process
In Figure 3 of the paper, it appears that generation was performed including protein tokens, but in the released code, generation seems to start from the <LIG_START> token. Could you clarify which approach was actually used?


## 2. Generation fine-tuning results (Vina score and Vina dock)
The paper reports both Vina score and Vina dock results. Were these obtained by first generating SMILES strings with the generation fine-tuned model, then docking them using the docking fine-tuned model to get coordinates and scores? Or were they derived directly from the generation fine-tuned model?


## 3. About generation task datasets
According to the paper and the code you provided on GitHub, my understanding is that during the generation finetuning process, 128 samples are generated at each step for 500 steps. Then, after the RL finetuning for each protein pocket is complete, the top 100 samples are selected for evaluation.

I have a few questions regarding this process:

The dataset used for RL finetuning is also included in the pre-training data. Is it appropriate to use this data for the final evaluation? I am concerned about potential data leakage.

In other papers, performance is typically compared by randomly sampling 100 molecules. In contrast, this paper appears to generate a very large number of samples and then selects only the top 100 for evaluation. I am wondering if this might be an unfair comparison.

I am also curious whether a separate, specialized model was trained for each individual protein pocket.


## 4. Evaluation code
Would it be possible to share the evaluation code used to compute the scores reported in the paper for both docking and generation fine-tuning? This does not appear to be included in the GitHub repository.


## 5. Docking test results
I followed the code provided in the GitHub repository exactly and created my own docking evaluation script based on the methodology described in your paper. However, my results are significantly different from those reported. Since the evaluation was performed on data that had already been through the pre-processing pipeline, I am unsure why there is such a large discrepancy.

This leads me to ask were there any additional pre-training or finetuning steps performed that differ from what is described in the paper or the public GitHub code?

Here are my experimental results:

| Metric | My Result | Result in Paper |
| :--- | :--- | :--- |
| **RMSD < 1.0 Å** | 0.80% | 43.8%
| **RMSD < 1.5 Å** | 4.78% |
| **RMSD < 2.0 Å** | 14.74% | 84.9% |
| **RMSD < 3.0 Å** | 34.26% | 96.4% |
| **RMSD < 5.0 Å** | 66.53% | 98.8% |
| **Average RMSD** | 4.52 | 1.29 |

As you can see, these values differ considerably from those in the paper.


## 6. Dataset repetition discrepancy
The paper mentions repeating the protein pocket dataset five times and the pocket-ligand dataset twenty times, whereas the github code sets the number of epochs to two and ten, respectively. Could you clarify which values were actually used?

Thank you very much for your time and for making the code publicly available. I would greatly appreciate any insights or materials you could share to help with faithfully reproducing your work.

Best regards.

---

> ### Public Comment · ~Xiuyuan_Hu1 · 2025-08-10
>
> Dear Nahyun,
>
> Thank you for your interest in our work. We sincerely appreciate your efforts to reproduce our results and your thoughtful questions.
>
> We acknowledge that there may be some typos or ambiguities in the released code, which could be misleading. We will carefully review and update the code as soon as possible to ensure clarity. Additionally, we are working to address the challenges of uploading large model checkpoints due to internet restrictions in mainland China, and we will do our best to make these available to facilitate your work.
>
> Regarding your **Point 5**, the discrepancy between your results and those reported in the paper is significant—far beyond what might be expected from experimental randomness. Given that your results are even lower than the performance of "w/o pretraining" (i.e., training on PDBbind alone), there may be a misalignment in the training process, possibly due to issues in our code. We deeply apologize for any confusion or inconvenience this has caused and will prioritize investigating this further.
>
> Thank you again for your patience and for bringing these issues to our attention. Please don’t hesitate to reach out if you have further questions in the meantime.
>
> Best regards,
> Xiuyuan Hu, on behalf of all authors

---

### Meta-Review · Area_Chair_wmVB · 2024-12-20

**Metareview:**

This submission received feedback from four reviewers, almost all of whom provided positive reviews. Even though Reviewer H1Mo scored 5 in the first round and with no further feedback after rebuttal, the authors provided replies point-by-point and solved most of the issues from my perspective. Therefore, my recommendation is 'Accept (poster).

**Additional Comments On Reviewer Discussion:**

This submission received feedback from four reviewers, almost all of whom provided positive reviews (one scored 8, two scored 6 and one scored 5). Even though Reviewer H1Mo scored 5 in the first round and with no further feedback after rebuttal, the authors provided replies point-by-point and solved most of the issues from my perspective.

---

### Decision · Program_Chairs · 2025-01-22

Accept (Poster)